# Combination of Coagulation–Flocculation–Decantation and Ozonation Processes for Winery Wastewater Treatment

**DOI:** 10.3390/ijerph18168882

**Published:** 2021-08-23

**Authors:** Nuno Jorge, Ana R. Teixeira, Carlos C. Matos, Marco S. Lucas, José A. Peres

**Affiliations:** 1Escuela Internacional de Doctorado (EIDO), Campus da Auga, Campus Universitário de Ourense, Universidade de Vigo, 32004 Ourense, Spain; njorge@uvigo.es; 2Centro de Química de Vila Real (CQVR), Departamento de Química, Universidade de Trás-os-Montes e Alto Douro (UTAD), Quinta de Prados, 5000-801 Vila Real, Portugal; ritamourateixeira@gmail.com (A.R.T.); cmatos@utad.pt (C.C.M.); mlucas@utad.pt (M.S.L.)

**Keywords:** coagulation–flocculation–decantation, germination index, ozone, potassium caseinate, UV-C radiation, winery wastewater

## Abstract

This research assessed a novel treatment process of winery wastewater, through the application of a chemical-based process aiming to decrease the high organic carbon content, which represents a difficulty for wastewater treatment plants and a public health problem. Firstly, a coagulation–flocculation–decantation process (CFD process) was optimized by a simplex lattice design. Afterwards, the efficiency of a UV-C/ferrous iron/ozone system was assessed for organic carbon removal in winery wastewater. This system was applied alone and in combination with the CFD process (as a pre- and post-treatment). The coagulation–flocculation–decantation process, with a mixture of 0.48 g/L potassium caseinate and 0.52 g/L bentonite at pH 4.0, achieved 98.3, 97.6, and 87.8% removals of turbidity, total suspended solids, and total polyphenols, respectively. For the ozonation process, the required pH and ferrous iron concentration (Fe^2+^) were crucial variables in treatment optimization. With the application of the best operational conditions (pH = 4.0, [Fe^2+^] = 1.0 mM), the UV-C/ferrous iron/ozone system achieved 63.2% total organic carbon (TOC) removal and an energy consumption of 1843 kWh∙m^−3^∙order^−1^. The combination of CFD and ozonation processes increased the TOC removal to 66.1 and 65.5%, respectively, for the ozone/ferrous iron/UV-C/CFD and CFD/ozone/ferrous iron/UV-C systems. In addition, the germination index of several seeds was assessed and excellent values (>80%) were observed, which revealed the reduction in phytotoxicity. In conclusion, the combination of CFD and UV-C/ferrous iron/ozone processes is efficient for WW treatment.

## 1. Introduction

Portugal is a Mediterranean wine producer, with an approximated vineyard area of 191,000 ha and a wine production value of 6.4 MhL, in 2020 [1]. This extensive wine production requires large amounts of water, to perform several activities that are necessary to ensure the quality of the wines, such as the floor and equipment washing, rinsing of the transfer lines, barrel cleaning, bottling facilities, filtration units, etc. [2,3,4]. The high consumption of water inevitably leads to the generation of large amounts of winery wastewaters (WW). The environmental impact of wastewater from the wine industry is highly detrimental if it is released without proper treatment, causing water pollution, soil degradation, and damage to the vegetation by odors and gaseous emissions [2,5]. Currently, the most widespread decontamination approaches are activated sludge reactors [6]; however, the seasonal character of these wastewaters makes it difficult for the microorganisms to adapt. In addition, during the vintage stage, the pollutant loads and wastewater volumes that are produced are higher, requiring longer retention times in the activated sludge reactors, which makes these biologic reactors oversized for most of the year [7].

The coagulation–flocculation–decantation process (CFD process) is one of the most mature and effective processes involved in wastewater treatment, which can remove the suspended particles and most of the colloid particles, by the formation of flocs. Generally, the CFD mechanisms can be categorized into the following four kinds: (1) simple charge neutralization, (2) charge patching, (3) bridging, and (4) sweeping [8,9]. The application of inorganic salt coagulants, such as iron and aluminum, has been reported in the treatment of winery wastewater [10], mature landfill leachate [11], cork processing wastewaters [12], among others. Nonetheless, there are several drawbacks that are associated with the use of these metallic salts, for instance, their high sensitivity to pH, their ineffectiveness on miniscule particles in low temperatures, and the production of large amounts of sludge containing metal hydroxides [13]. In this work, the application of a mixture of potassium caseinate, activated sodium bentonite, and polyvinylpolypyrrolidone was tested, as an alternative to metallic coagulants, which, to the best of our knowledge, have not been applied to the treatment of winery wastewater.

Another alternative to biologic treatments are advanced oxidation processes (AOPs), which are based on the production of hydroxyl radicals (${\mathrm{HO}}^{•}$) and are capable of promoting, in a non-selective manner, both the degradation and mineralization of pollutants into CO_2_, H_2_O, and inorganic salts [14,15]. Among the different AOPs, the following have been reported to be used in the treatment of winery wastewater: the application of the homogeneous Fenton process [16], homogeneous photo-Fenton process [17,18], sulfate radicals [19,20], heterogeneous photo-Fenton process [21,22], and ozonation process [23]. In this work, the application of ozonation, as a complement to the CFD process in the treatment of winery wastewater, was studied. Ozone is an unstable gas, with a characteristically penetrating odor and partial solubility in water. In wastewater treatment, the advantages of ozone are evident, since it is a powerful oxidant, with a redox potential of 2.07 V in alkaline solution, making it able to oxidize several inorganic and organic substances, when compared to other oxidizing agents, such as H_2_O_2_ (1.77 V), ${\mathrm{HO}}_{2}^{•}$ (1.70 V), Cl_2_ (1.09 V), and O_2_ (0.40 V) [24]. In addition, generated H_2_O_2_ can considerably enhance the ${\mathrm{HO}}^{•}$ formation from O_3_ decomposition, during the O_3_/UV process when compared to conventional ozonation [25,26]. It was observed by several authors that ozone, in combination with UV-C radiation, was effective in the treatment of wastewater with a high polyphenol content, such as that found in the cork manufacturing industry [27], the olive oil industry [28], and the wine distillery industry [29]. In this work, the application of the system O_3_/Fe^2+^/UV-C was tested, which, to our knowledge, has not been hitherto performed in winery wastewater treatment and, therefore, its effects in organic carbon reduction are still unknown.

In order to answer these important questions, the aim of this work is (1) to perform the treatment of winery wastewater using CFD, with the application of potassium caseinate, activated sodium bentonite, and PVPP; (2) to optimize the coagulant mixture by the performance of a simplex lattice design; (3) to optimize the ozonation process; (4) to evaluate the efficiency of the combined CFD/ozonation processes; and, finally, (5) to study the effect of the combined CFD/ozonation processes in the phytotoxicity reduction in plant seeds, and changes in the phenolic and chromatic characteristics of the wastewater.

## 2. Materials and Methods

### 2.1. Reagents and Winery Wastewater Sampling

Winery wastewater was collected from a cellar located in the Douro region (Northern Portugal). This agroindustry is a private company dedicated to the production of table wine. It is responsible for receiving grapes and for their processing, from crushing, must fermentation, wine stabilization and filtration, and finally bottling. After collecting the samples in plastic containers to be transported to the laboratory, they were stored at −40 °C. This work was performed at the University of Trás-os-Montes and Alto Douro, located in Vila Real, Portugal, latitude 41°17′9.18′′ N and longitude 7°44′21.45′′ W.

Activated sodium bentonite was purchased from Angelo Coimbra & Ca., Lda, Maia, Portugal, potassium caseinate and polyvinylpolypyrrolidone (PVPP) from A. Freitas Vilar, Lisboa, Portugal, and ferrous sulphate heptahydrate (FeSO_4_•7H_2_O) from Panreac, Barcelona, Spain. NaOH and H_2_SO_4_ (95%) were both obtained from Analar Normapur, Vila Nova de Gaia, Portugal. Deionized water was used to prepare the respective solutions.

### 2.2. Analytical Techniques

Different physical–chemical parameters were monitored in order to characterize the WW, including the chemical oxygen demand (COD), the biochemical oxygen demand (BOD_5_), the total organic carbon (TOC) and the total polyphenols. The main chemical parameters measured are shown in Table 1. The COD and BOD_5_ were determined according to Standard Methods (5220D; 5210D; respectively) [30]. COD analysis was carried out in a COD reactor from HACH Co. (Loveland, CO, USA), and a HACH DR 2400 spectrophotometer (Loveland, CO, USA) was used for colorimetric measurement. Biochemical oxygen demand (BOD_5_) was determined using a respirometric OxiTop system. The turbidity was determined by a 2100N IS turbidimeter (Hach, Loveland, CO, USA), pH by a 3510 pH meter (Jenway, Cole-Parmer, UK) and conductivity by a portable condutivimeter, VWR C030 (VWR, V. Nova de Gaia, Portugal). These measurements were determined in accordance to the methodology of the Standard Methods [30]. The TOC content (mg C/L) was determined using a Shimadzu TOC-L CSH analyzer (Shimadzu, Kyoto, Japan). Total polyphenols were evaluated following the Folin–Ciocalteu method [31]. Dissolved ozone was measured by application of the AccuVac Ampul procedure (Ozone AccuVac^®^ Ampules, 0–1.5 mg/L, HACH, Loveland, CO, USA). The ferrous iron concentrations was analyzed by atomic absorption spectroscopy (AAS) using a Thermo Scientific iCE 3000 SERIES (Thermo Fisher Scientific, Waltham, MA, USA).

Phytotoxicity tests were performed by germination of onion, cucumber, lettuce and corn seeds (standard species recommended by the US Environmental Protection Agency, the US Food and Drug Administration, and the Organization for Economic Cooperation and Development [32]) and determined by Equation (1) in accordance to Varnero et al. [33] and Tiquia et Tam [34], as follows:(1)$$\mathrm{GI}\left(\%\right)=\frac{{\bar{N}}_{SG,T}}{{\bar{N}}_{SG,B}}*\frac{{\bar{L}}_{R,T}}{{\bar{L}}_{R,B}}*100$$
where GI is the germination index, ${\bar{N}}_{SG,T}$ is the arithmetic mean of the number of germinated seeds in each extract (wastewater), ${\bar{N}}_{SG,B}$ is the arithmetic mean of the number of germinated seeds on standard solution (distilled water), ${\bar{L}}_{R,T}$ is the mean root length in each extract (wastewater) and ${\bar{L}}_{R,B}$ is the mean root length in control (distilled water).

Let Xi be defined as the removal of a given indicator (%) of water contamination (turbidity, TSS, TOC, COD and total polyphenols) that is achieved by a treatment (Equation (2)) [12], as follows:(2)$$X_{i}\left(\%\right)=\frac{C_{0}{-C}_{f}}{C_{0}}*100$$
where C_0_ and C_f_ are the initial and final concentrations, respectively, of parameter i.

### 2.3. Phenolic and Chromatic (CIELab) Characterization

Color intensity (CI) and hue were determined by the OIV method [35], total polyphenol index (TPI) was determined by Curvelo-Garcia method [36], total phenols, non-flavonoids and flavonoids were determined according to Kramling and Singleton [37]. Total anthocyanins were analyzed by SO_2_ bleaching method, described by Ribéreau-Gayon et al., [38], colored anthocyanins (*CA*), total pigments (*T**P*) and polymeric pigments (*PP*) were analyzed by the method described by Somers and Evans [39], and total tannins were determined by the LA method [40]. All samples were analyzed by a spectrophotometer (GENESYS^TM^ 10 series spectrophotometers). The absorption spectra of WW samples were recorded with a Shimadzu UV-2101 spectrophotometer (Shimadzu, Kyoto, Japan) scanned from a range between 380 and 770 nm, with 5 nm distance, using 1 cm path length quartz cells. Data were collected to determine *L* (lightness), *a* (redness), and *b* (yellowness) coordinates using the CIELab 1976 method. This allows reliable quantification of the overall color difference of a sample when compared to a reference sample (blank). Color differences can be distinguished by the human eye when the differences between $\Delta E_{ab}^{*}$ values are greater than two units, in accordance to Spagna et al. [41]. All analyses were performed in duplicate. Table 2 resumes the formulas used in this work.

### 2.4. Coagulants Characterization

The FTIR spectra were obtained by mixing 2 mg of coagulants with 200 mg KBr. The powder mixtures were then inserted into molds and pressed at 10 ton/cm^2^ to obtain the transparent pellets. The samples were analyzed with a Bruker Tensor 27 spectrometer and the infrared spectra in transmission mode was recorded in the 4000–400 cm^−1^ frequency region. The microstructural characterization was carried out with a scanning electron microscop (FEI QUANTA 400 SEM/ESEM, Fei Quanta, Hillsboro, WA, USA) and the chemical composition of the different catalysts was estimated (Table 3) using the energy-dispersive X-ray spectroscopy (EDS/EDAX, PAN’alytical X’Pert PRO, Davis, CA, USA).

The textural parameters of samples were obtained from N_2_ adsorption–desorption isotherms at 77 K using a Micromeritics ASAP 2020 apparatus (TriStar II Plus, Micromeritics Instrument Corporation, Norcross, GA, USA). The samples were degassed at 150 °C up to 10^−4^ Torr before analysis. The specific surface area (SBET) was determined by applying the Gurevitsch’s rule at a relative pressure p/p_0_ = 0.30 and according to the Brunauer, Emmet, Teller (BET) method from the linear part of the nitrogen adsorption isotherms. Different pore volumes were determined by the Barrett, Joyner, Halenda model (BJH model).

From FTIR analysis (Figure 1), potassium caseinate shows a sharp peak at hydrophilic O–H stretching at 3523 cm^−1^ and a strong vibration of hydrophobic C–H stretching (proteins) at 2962 cm^−1^. Further, a –C=O stretching vibration (amide I) was observed at 1643 cm^−1^, C–N stretching and N–H bending at 1531 cm^−1^ (characteristic amide II band), a C–H bending deformation and –CH_3_ symmetrical deformation at 1446 and 1386 cm^−1^, respectively, and –C–NH_2_ stretching (amide III) at 1240 cm^−1^ [43,44].

PVPP shows an O–H stretching assigned to the stretching vibration of hydroxyl group (OH) at 3481 cm^−1^ [45], CH_2_ asymmetric stretching vibration (ring) at 2956 cm^−1^ [46], C–H stretching at 2895 cm^−1^ [45], C=O stretching of PVPP at 1651 cm^−1^ [47,48,49,50], CH_2_ scissoring vibrations at 1463 cm^−1^ [46], C–N stretching or C–O stretching at 1288 cm^−1^ [51], twisting of CH_2_ of PVPP at 1228 cm^−1^ and CH_2_ rocking at 1020 cm^−1^ [45,46].

The activated sodium bentonite shows the stretching vibration of structural O–H groups at 3645 cm^−1^, structural Si–O groups at 1103, 999 and 789 cm^−1^, structural Al–Al–OH groups at 902 cm^−1^, structural Al–Fe–OH groups at 883 cm^−1^, the free and interlayer water in bond stretching vibration at 3396 cm^−1^, and adsorbed water-yielded bending at 1643 cm^−1^ [52,53,54,55].

The results obtained by BET analysis (Table 4) showed that bentonite exhibited a mesoporous structure with a specific surface area of 8.8 m^2^/g, a total pore volume of 0.045 cm^3^/g and a particle size of 4.0 nm. The respective isotherms can be classified as type II, where unrestricted monolayer–multilayer adsorption occurs, and the behavior of the hysteresis loops can be associated with type H3, which usually corresponds to aggregates of plate-like particles forming slit-like pores [56], which is in agreement with these material structures. Potassium caseinate had a specific surface area of 1.0 m^2^/g, PVPP’s surface area was not quantifiable. Total pore volume and particle size were not quantifiable for potassium caseinate and PVPP, which indicates that these coagulants have a microporous structure. The shape of its N_2_ adsorption–desorption isotherm was a type I isotherm, typical of microporous solids with relatively small external surfaces, as defined by the International Union of Pure and Applied Chemistry (IUPAC) [57].

### 2.5. Coagulation–Flocculatuion–Decantation Experiments

CFD experiments were performed in a jar-test apparatus (ISCO JF-4). Several trials were performed using 500 mL of effluent in 1000 mL beakers. Fixed conditions were set as the following: pH 4.0, rapid mixing 150 rpm/3 min, slow mixing 20 rpm/20 min, temperature 298 K, sedimentation period 12 h. The CFD experiments were developed with the statistical software Minitab 18.0 (State College, Pennsylvania, USA), and applied the simplex lattice design (SLD) as follows:Application of bentonite, potassium caseinate and PVPP with a lower and upper dosage of 0 and 1.0 g/L, respectively;Application of maximum mixture dosage of 1.0 g/L.

All the experiments were performed in triplicate and the observed standard deviation was always less than 5% of the reported values. Statistical analysis was performed by OriginLab 2019 software (Northampton, MA, USA).

### 2.6. Ozonation Experiments

Batch experiments for the ozonation process were performed using an air pump IDEA-R AP2 (1.8 W/1000 ccO_2_/min) from SICCE. Oxygen was converted into ozone by an ozone generator 1KNT-24 (25 W) of 1000BT-12 with low working noise (<30 DB). This ozone generator uses high-output corona discharge ozone tube, together with a high Óow ball bearing cooling fan (48 CFM) to ensure steady ozone output (up to 300 mg/hr). Figure 2 shows the setup of the ozone reactor used in this work.

The photoreactor was fitted with a Heraeus TNN 15/32 lamp (14.5 cm in length and 2.5 cm in diameter), mounted in the axial position inside the reactor. The spectral output of the low-pressure mercury vapor lamp emitted mainly (85–90%) at 253.7 nm and about 7–10% at 184.9 nm. The experiments were carried out as follows:Performance of different pH conditions (4.0, 7.0, 9.0 and 11.0) under the following operational conditions: [Fe^2+^] = 1.0 mM, ozone flow rate 5 mg/min, air flow 1.0 L/min, agitation 350 rpm and UV-C mercury lamp (254 nm);Performance of different Fe^2+^ concentrations (0.0, 0.5, 1.0 and 2.0) at pH 4.0, ozone flow rate 5 mg/min, air flow 1.0 L/min, agitation 350 rpm and a UV-C mercury lamp (254 nm).

All the experiments were performed in triplicate and the observed standard deviation was always less than 5% of the reported values. Statistical analysis was performed with OriginLab 2019 software (Northampton, MA, USA).

## 3. Results and Discussion

### 3.1. Coagulation–Flocculation–Decantation Experiments

#### 3.1.1. Simplex Lattice Design—Model Establishment

In Table 1, it is observed that the WW has high levels of turbidity (1040 NTU), total suspended solids (TSS) (2430 mg/L), and organic content (1962 mg C/L). The CFD process is essential to reduce these parameters, to improve the ozonation treatment efficiency, since the presence of refractory compounds is considered to be hydroxyl radical scavengers. In addition, without suspended solids, light can better penetrate through the solution, to trigger ozone decomposition [11]. The CFD experiments were performed by a simplex lattice design (SLD), in which seven different tests were performed, to optimize the combination of coagulants. The experimental and predicted values obtained after the assessment of turbidity, TSS, chemical oxygen demand (COD), and total organic carbon (TOC), are listed in Table 5.

The regression models of the four responses were established by linear regression fitting (Equations (3)–(6)), as follows:Y_1_ (Turbidity) = 99.5038X_1_ + 99.6038X_2_ + 98.9038X_3_ + 0.391X_1_X_2_ − 0.209X_1_X_3_ + 1.591X_2_X_3_; R^2^ = 99.5%; R^2^ adjusted = 97.3%.(3)
Y_2_ (TSS) = 97.9083X_1_ + 98.3083X_2_ + 97.5083X_3_ + 0.10X_1_X_2_ + 0.10X_1_X_3_ + 1.30X_2_X_3_; R^2^ = 97.9%; R^2^ adjusted = 87.7%.(4)
Y_3_ (COD) = 48.61X_1_ + 54.41X_2_ + 56.31X_3_ + 19.8X_1_X_2_ + 3.6X_1_X_3_ − 36.0X_2_X_3_; R^2^ = 95.5%; R^2^ adjusted = 73.1%.(5)
Y_4_ (TOC) = 31.635X_1_ + 28.435X_2_ + 37.035X_3_ + 57.44X_1_X_2_ − 51.76X_1_X_3_ − 9.76X_2_X_3_; R^2^ = 99.7%; R^2^ adjusted = 98.1%.(6)

Validation of the statistical design model is a very important parameter, in order to assess the relevance of the obtained results. Univariate analysis of variance (ANOVA) provides an extremely powerful and useful tool for the statistical tests of different factors and their interactions in experiments [58]. The regression-adjusted average squares and linear regression-adjusted average squares, allowed the calculation of the Fisher ratios (F-value), which allowed the determination of the statistical significance (*p*-value). Appendix A (see Appendix A) shows that the P-value for regression, linear regression and quadratic regression of turbidity, TSS, COD, and TOC, is non-significant for a *p* < 0.05 [59].

To establish the reliability of the results that were obtained by the SLD statistic design, it was necessary to evaluate other sets of parameters. By the performance of linear fit, the regression equations were obtained. These equations established a relation between the observed results and the predicted results, which were evaluated by the calculation of R^2^ and R^2^ adjusted. By observation of the R^2^ values, turbidity, TSS, COD, and TOC had values of 99.5, 97.9, 95.5, and 99.7%, respectively, which were greater than 80.0%, ensuring a very good model fit [60,61]. The R^2^ adjusted that was obtained from the turbidity, TSS, COD, and TOC regression equations (97.3, 87.7, 73.1, and 98.1%, respectively), was also high, indicating the strong significance of the model [61,62].

The adequacy of the model was also evaluated by use of diagnostic plots, such as a normal probability plot of the standardized residuals, and the plot between individual residual values and the fitted values (Figure 3). The normal probability plot shows the distribution of the residual value, which is defined as the difference between the predicted (model) and observed (experimental) values. From the normal probability plot, the population is claimed not to be normally distributed if, and only if, the n points do not fall close to a straight line [63]. 

In Figure 3, the formation of a straight line was observed and the residual values were normally distributed on both sides of the line, which indicated that the experimental points of turbidity, TSS, COD, and TOC were reasonably aligned with the predicted values.

After analyzing the plots between individual residual values and the fitted values, it was observed that the turbidity, TSS, COD, and TOC residual values were scattered randomly around zero, which was similar to Harbi et al. [64], who observed a uniform scattering of the residual values after the performance of an SLD statistic design on a PCR detection method for nitrite reductase genes.

#### 3.1.2. Simplex Lattice Design—Model Optimization

To improve the CFD optimization, an SLD statistic design was performed, in which three different coagulants, potassium caseinate (X_1_), activated sodium bentonite (X_2_), and PVPP (X_3_), were tested under seven different combinations, with the evaluation of turbidity, TSS, COD, and TOC (Table 5), under the following operational conditions: pH 4.0, rapid mixing 150 rpm/3 min, slow mixing 20 rpm/20 min, temperature 298 K, and sedimentation period 12 h. The maximum dosage (1.0 g/L) and pH 4.0 were selected, based on the works of Cosme et al. [65,66], who performed coagulation with these coagulants on the natural pH (3.3–3.6) of wine. Additionally, potassium caseinate has an isoelectric point of 4.6 [67] and, at a pH higher than 4.6, potassium caseinate is dissolved. The activated sodium bentonite that was used in this work presents a pH of 7.4, which corresponds to its isoelectric point (pH = 7); however, bentonite becomes electropositive at pH 3.6–4.0 [68] and, therefore, after bentonite addition to the wastewater, its pH was corrected to 4.0. The variable charge of clays is affected by the pH, due to the ionization of its external hydroxyl groups. Therefore, the external sites may acquire positive (${\mathrm{OH}}_{2}^{+}$) or negative charge (O−−), whether the pH is lower or higher than its isoelectric point (pH at the point of zero charge) [69,70]. In the work of Guimarães et al. [69], when the clay was mixed with WW at pH 4.0, a mechanism of adsorption occurred, in which a high concentration of amphoteric flavylium species were adsorbed by the clay. Therefore, pH 4.0 is the best pH for the performance of the CFD process, with activated sodium bentonite. The PVPP was tested in this work, based on the research of Labord et al. [71], in which it was reported that PVPP had a high affinity for polyphenols, according to tests conducted in wine treatment. PVPP is a high-molecular-weight polymer, which is insoluble in water [38]. It interacts with polyphenols via H bonds between their CO-N linkages and phenol groups [72].

Figure 4 shows the interactions of activated sodium bentonite, potassium caseinate, and PVPP, with the colloids present in the wastewater. When bentonite was added, a mechanism of adsorption and charge neutralization occurred, attracting the colloids present in the WW to the interlayer region of the clay [73]. A similar mechanism was observed with the employment of potassium caseinate, which due to the different charges, attracted the negatively chargeed particles to the positively charged proteins, producing heavy aggregated particles, which precipitate by gravity [74]. PVPP acted by a mechanism of adsorption and interparticle bridging. The polymer chains of PVPP adsorbed the colloids from the WW, as a result of (1) coulombic (charge–charge) interactions, (2) dipole interaction, (3) hydrogen bonding, and (4) van der Waals forces of attraction [75]. In addition, PVPP creates a tridimensional net that adsorbs on the available surface sites of other particles (such as bentonite and potassium caseinate), thus creating a ‘‘bridge’’ between the particles surfaces, resulting in larger particles that settle more efficiently by gravity.

The statistical analysis that was obtained from the SLD design was converted into an optimization chart, which related the best results that were obtained from the CFD process. In accordance with the optimization chart (Appendix A), the mixture of 0.48 g/L potassium caseinate (X_1_) and 0.52 g/L bentonite (X_2_) could reach a maximum removal of COD, TOC, turbidity, and TSS of 56.5, 44.3, 99.6, and 98.1%, respectively. Therefore, the optimal conditions that were selected for the CFD process were as follows: 0.48 g/L potassium caseinate, 0.52 g/L bentonite, pH 4.0, temperature 298 K, rapid mix 150 rpm/3 min, slow mix 20 rpm/20 min, and sedimentation time 12 h. After 12 h of sedimentation (Appendix A, Figure 5), a 98.3, 97.6, 48.0, and 44.6% removals of turbidity, TSS, COD, and TOC were observed, which are similar to the values predicted by the optimization chart in Appendix A. Considering the Portuguese Decree Law nº 236/98 for residual water discharge, it was observed that TSS achieved the legal value (60 mg/L).

### 3.2. Ozonation Experiments

#### 3.2.1. Effect of pH

In Section 3.1, the CFD process was optimized by employing an SLD statistical design. However, the CFD process was insufficient to substantially decrease the high organic carbon content that was present in the WW, thus, the application of a further chemical oxidation process, based on the reaction between ozone and a catalyst (Fe^2+^), under UV-C radiation, was necessary. In the work of Huang et al. [76], it was observed that a combination of iron with O_3_ reached higher dissolved organic carbon (DOC) removal (53%), regarding O_3_ alone (32%), in the treatment of pharmaceutical wastewater. However, to our knowledge, the O_3_/Fe^2+^/UV-C process was never applied to the treatment of WW. Therefore, the O_3_/Fe^2+^/UV-C process was optimized, to act as a CFD complementary process. In order to maximize TOC removal, different pH’s (4.0, 7.0, 9.0, and 11.0) were tested, under the following operational conditions: [Fe^2+^] = 1.0 mM, ozone flow rate 5 mg/min, air flow 1.0 L/min, agitation 350 rpm, time 600 min, and a UV-C mercury lamp (254 nm). In Figure 6, the O_3_/Fe^2+^/UV-C system is represented in closed symbols, while the blank experiments, at pH 4.0 (O_3_, UV-C, and O_3_/UV-C), are represented in open symbols. The effect of the pH in the blank experiments is shown in Appendix A. It was observed that with different initial pH (4.0, 7.0, 9.0, and 11.0), there was a TOC removal of 31.9, 28.6, 11.2, and 35.8%, respectively, for O_3_; 33.1, 0.9, 0.0, and 4.1%, respectively, for UV-C; and 57.5, 37.6, 36.5, and 38.0%, respectively, for O_3_/UV-C. These results were in agreement with the work of Hassanshahi and Karimi-Jashni [77], which observed that the pH had little effect in the COD removal of gray water, by the ozonation process.

The results in Figure 6 showed a lower TOC removal efficiency for a single O_3_ application, regarding O_3_/UV-C. These results can be explained by two mechanisms through which ozone can degrade organic pollutants, such as direct electrophilic attack (Equation (7)) and indirect attack, through the formation of hydroxyl radicals (Equations (8) and (9)) [78,79], which are shown as follows:O_3_ + R → R_OX_(7)
(8)$$O_{3}\;+\;{\mathrm{HO}}^{-}\;\to \;{\mathrm{HO}}^{•}\;+\;(O_{2}^{•}\;↔\;{\mathrm{HO}}_{2}^{•})$$
(9)$${\mathrm{HO}}^{•}\;+R^{•}\;\to \;R{’}_{\mathrm{OX}}$$
where R is the organic solutes and Rox is the oxidized organic products.

With the application of UV-C radiation at 254 nm, in combination with O_3_, the TOC removal increased, because ozone absorbs UV-C light [80,81] and the formation of H_2_O_2_, by ozone photolysis occurs, which, in turn, produced hydroxyl radicals (${\mathrm{HO}}^{•}$), as observed in Equations (10)–(14) [82] as follows:O_3_ + H_2_O + *hv* → H_2_O_2_ + O_2_(10)
(11)$$H_{2}O_{2}\;+\;hv\;\to \;2{\mathrm{HO}}^{•}$$
(12)$$H_{2}O_{2}\;↔\;{\mathrm{HO}}_{2}^{-}\;+\;H^{+}$$
(13)$$O_{3}\;+\;{\mathrm{HO}}_{2}^{-}\;\to \;O_{3}^{•-}\;+\;{\mathrm{HO}}_{2}^{•}$$
(14)$$O_{3}^{•-}\;+\;H^{+}\;\to \;{\mathrm{HO}}_{3}^{•}\;\to \;{\mathrm{HO}}^{•}\;+\;O_{2}$$

Agustina et al. [79] also observed two more reactions that occurred under O_3_/UV-C, as described in Equations (15) and (16) as follows:(15)$${\mathrm{HO}}_{2}^{•}\;\to \;O_{2}^{•-}\;+\;H^{+}$$
(16)$$O_{3}\;+\;O_{2}^{•-}\;\to \;O_{3}^{•-}\;+\;O_{2}$$

With the application of the O_3_/Fe^2+^/UV-C system, a TOC removal of 63.2%, 42.4%, 52.7%, and 49.6%, respectively, was observed, for pH 4.0, 7.0, 9.0, and 11.0. In Equation (10), it was observed that the photolysis of O_3_ could produce H_2_O_2_. When Fe^2+^ was added, the catalyst decomposed the H_2_O_2_, producing ${\mathrm{HO}}^{•}$ radicals (Equations (17)–(19)) [83,84], thus increasing the rate of TOC removal. With the application of UV-C radiation, the regeneration of Fe^3+^ to Fe^2+^ (Equation (20)) took place [85,86], explaing the high efficiency of the system.
(17)$${\mathrm{Fe}}^{2+}\;+\;H_{2}O_{2}\;\to \;{\mathrm{HO}}^{•}+{\mathrm{Fe}}^{3+}+{\mathrm{HO}}^{-}$$
(18)$${\mathrm{Fe}}^{3+}\;+\;H_{2}O_{2}\;\to \;{\mathrm{FeHO}}_{2}^{2+}$$
(19)$${\mathrm{FeHO}}_{2}^{2+}\;\to \;{\mathrm{HO}}_{2}^{•}+{\mathrm{Fe}}^{2+}$$
(20)$${\mathrm{Fe}}^{3+}\;+\;H_{2}O_{2}\;+\;hv\;\to \;{\mathrm{HO}}^{•}+{\mathrm{Fe}}^{2+}+H^{+}$$

The TOC removal results fitted a pseudo-first-order kinetic rate (Equation (21)) [87].
(21)$$\ln\frac{{\left[\mathrm{TOC}\right]}_{t}}{{\left[\mathrm{TOC}\right]}_{0}}=-\mathrm{kt}$$
where ${\left[\mathrm{TOC}\right]}_{0}$ and ${\left[\mathrm{TOC}\right]}_{t}$ are the TOC concentrations at time 0 and t in mg C/L.

As the O_3_/Fe^2+^/UV-C reaction proceeds, the concentration of TOC decreases. Another measure of the rate of a reaction, relating concentration to time, is the half-life, $t_{1/2}$, which is the time required for the concentration of TOC to decrease to half of its initial concentration. We can obtain an expression for $t_{1/2}$, for a first-order reaction, as described by Equation (22) [87], as follows:(22)$$t_{1/2}=\frac{0.693}{k}$$

It was observed that $k_{\mathrm{pH}\;4.0}$ = 1.67 × 10^−3^ min^−1^ > $k_{\mathrm{pH}\;9.0}$ = 1.18 × 10^−3^ min^−1^ > $k_{\mathrm{pH}\;11}$ = 1.07 × 10^−3^ min^−1^ > $k_{\mathrm{pH}\;7.0}$ = 9.41 × 10^−4^ min^−1^. The decrease in the kinetic rate, with the increase in pH > 4.0, could be explained by the precipitation of iron at alkaline pH, in the form of iron hydroxide ([Fe^2+^] leached = 12.6, 35.4, 42.6, and 188.8 mg Fe/L, respectively, for pH 4.0, 7.0, 9.0, and 11), which decreases the conversion of H_2_O_2_ to ${\mathrm{HO}}^{•}$ and reduces the transmission of the radiation [88]. Therefore, based in these results, pH 4.0 was selected as the best pH for the O_3_/Fe^2+^/UV-C process studied.

#### 3.2.2. Effect of Fe^2+^ Concentration

In the previous section, as the pH had a great influence on the removal of TOC from the WW, by the ozonation process, was described. It was also observed that the treatment using the combination of O_3_/UV-C generated H_2_O_2,_ which interacted with Fe^2+^ to produce ${\mathrm{HO}}^{•}$ radicals. Therefore, in this section, the Fe^2+^ concentration was varied (0.5–2.0 mM) under the following operational conditions: pH 4.0, ozone flow rate 5 mg/min, air flow 1.0 L/min, agitation 350 rpm, and radiation UV-C mercury lamp (254 nm).

In Figure 7, a TOC removal of 59.0, 63.2, and 66.7% (Appendix A), was observed, for 0.5, 1.0, and 2.0 mM Fe^2+^, respectively. These results agreed with Quiroz et al., [89], who stated that the application of iron with ozone improved the COD removal of industrial wastewater. In addition, in the work of Piera et al. [82], it was reported that Fe^2+^ can interact with O_3_ and produce ${\mathrm{HO}}^{•}$ radicals (Equations (23) and (24)), as follows:(23)$${\mathrm{Fe}}^{2+}+O_{3}\;\to \;{\mathrm{FeO}}^{2+}+O_{2}$$
(24)$${\mathrm{FeO}}^{2+}+H_{2}O\;\to \;{\mathrm{Fe}}^{3+}+{\mathrm{HO}}^{•}+{\mathrm{HO}}^{-}$$

After 600 min of reaction, an Fe^2+^ concentration of 1.84, 12.60, and 41.92 mg Fe/L, respectively, was observed, for 0.5, 1.0, and 2.0 mM. With the application of 0.5 mM Fe^2+^, there was a low concentration of Fe^2+^ present in the solution, to react with H_2_O_2_. The use of higher concentrations of ferrous iron resulted in a higher TOC removal, although the increase from 1.0 to 2.0 mM Fe^2+^ was very mild, possibly because scavenging reactions, between Fe^2+^ and ${\mathrm{HO}}^{•}$ radicals, may have occurred, as observed in Equations (25)–(26) [19], as follows: (25)$${\mathrm{Fe}}^{2+}\;+\;{\mathrm{HO}}^{•}\;\to \;{\mathrm{Fe}}^{3+}\;+\;{\mathrm{HO}}^{-}$$
(26)$${\mathrm{Fe}}^{2+}+{\mathrm{HO}}_{2}^{•}\;\to \;{\mathrm{Fe}}^{3+}\;+\;{\mathrm{HO}}_{2}^{-}$$

The results were fitted into a pseudo-first-order kinetic rate, and the following order was observed: $k_{2.0\;\mathrm{mM}}$ = 1.72 × 10^−3^ min^−1^ > $k_{1.0\;\mathrm{mM}}$ = 1.67 × 10^−3^ min^−1^ > $k_{0.5\;\mathrm{mM}}$ = 1.34 × 10^−3^ min^−1^. Due to the low differences in the kinetic rate that was observed between 1.0 and 2.0 mM Fe^2+^, and due to the high costs that are associated with the application of 2.0 mM Fe^2+^, a ferrous iron concentration of 1.0 mM was selected as the best concentration for the O_3_/Fe^2+^/UV-C process.

Bolton et al. [90] proposed figures of merit for electric-driven photocatalysis. For first-order kinetics, they proposed the electric energy per order ($E_{\mathrm{EO}}$) as the electric energy, in kWh that are required to reduce the concentration of a pollutant concentration C by one order of magnitude, according to Equation (27), where P is the rated power [kW] of the AOP system, V is the volume [L] of water or air treated in the time *t* [h], and TOC_i_ and TOC_f_ are the concentrations of total organic carbon at initial and *t* times.
(27)$$E_{\mathrm{EO}}=\frac{P*t*1000}{V*\log(\frac{{\mathrm{TOC}}_{i}}{{\mathrm{TOC}}_{f}})}\;\mathrm{Batch}\;\mathrm{operation}$$

The results are presented in Table 6 and, as expected by increasing the Fe^2+^ concentration, the $E_{\mathrm{EO}}$ values decreased. Lower $E_{\mathrm{EO}}$ values (in kWh∙m^−3^∙order^−1^) correspond to higher removal efficiencies, in terms of electrical power consumption [90]. The application of UV-C radiation alone achieved the lowest $E_{\mathrm{EO}}$ value (1720 kWh∙m^−3^∙order^−1^) in comparison to the O_3_/UV-C and O_3_ alone (2153 and 2996 kWh m^−3^∙order^−1^, respectively), due to the low power of the UV mercury lamp. However, the reaction kinetics of the UV-C alone, were much lower ($k_{\mathrm{UV}-C}$ = 7.17 × 10^−4^ min^−1^) in comparison to the other treatments, thus more energy will be required to achieve similar TOC degradation. With the application of the O_3_/Fe^2+^/UV-C system, the results in Table 6 clearly indicated that with 0.5 mM Fe^2+^_,_ the reaction took too much time, increasing the electric power consumption. However, little differences in the $E_{\mathrm{EO}}$ values were observed when 1.0 and 2.0 mM Fe^2+^ were applied, and, considering the high iron leaching that was observed with 2.0 mM Fe^2+^, the application of 1.0 mM Fe^2+^ becomes the best choice.

The comparative values of $E_{\mathrm{EO}},$ for the degradation of active pharmaceutical ingredients, real textile wastewater, and organic pollutants, by the ozonation process, are displayed in Table 7. In addition, a comparative study is also shown, in which the winery wastewater was treated by a UV-C/PMS/Co(II) system. The results showed that the winery wastewater is very difficult to treat (1843 kWh m^−3^∙order^−1^), with higher energy requirements, regarding the treatment of active pharmaceutical ingredients, real textile wastewater, and organic pollutants. These results were in agreement with Rodriguez-Chueca et al. [19], who observed a high $E_{\mathrm{EO}}$ for the treatment of WW (173 kWh∙m^−3^∙order^−1^). The application of ozonation processes in the treatment of WW is insufficient, with scarce examples of organic matter removal and energy consumption. Most of the authors studied the degradation of emerging contaminants and textile dye removal [91,92,93]; however, the degradation of the organic matter load of winery wastewater requires more demanding treatments, with higher energy consumption [20].

#### 3.2.3. Evaluation of Ozone Consumption

In the previous sections, we described how to optimize the ozonation process, and it was observed that under the best operational conditions, pH 4.0, [Fe^2+^] = 1.0 mM, ozone flow rate 5 mg/min, air flow 1.0 L/min, agitation 350 rpm and UV-C radiation (254 nm), a TOC removal of 63.2% was achieved. However, to understand the efficiency of the O_3_/Fe^2+^/UV-C process, the concentration of ozone dissolved in the wastewater was assessed (Appendix A). The rate of ozone consumption was monitored by AccuVac Ampul procedure, and the ozone that was lost in the process was measured by the difference between the ozone mass flow rate at the inlet of the reactor and the ozone dissolved in the WW.

In Figure 8, it was observed that under a constant injection of 5 mg O_3_/L, only 0.165, 0.330, 0.190, 0.270, and 0.360 mg O_3_/min reacted with the Fe^2+^, to produce hydroxyl radicals (${\mathrm{HO}}^{•}$), respectively, at time of 120, 240, 360, 480, and 600 min^−1^. This concentration of dissolved O_3_ was observed to be much lower than the concentration that was observed in the work of Lucas et al. [23], considering that the air was supplied by a small air pump, but a high TOC removal kinetic rate was observed with this process.

### 3.3. Combination of Coagulation–Flocculation–Decantation with Ozonation Processes

The WW is very toxic when discharged into the environment without proper treatment, due to the high levels of organic carbon that are present in its composition (Table 1). Previous treatments, with CFD and O_3_/Fe^2+^/UV-C, achieved a high TOC removal; however, both treatments showed limitations and, therefore, in this section, the processes of CFD and O_3_ were combined CFD/O_3_ and O_3_/CFD processes, with the application of the CFD process’ best operational conditions—0.48 g/L potassium caseinate, 0.52 g/L bentonite, pH 4.0, temperature 298 K, rapid mix 150 rpm/3 min, slow mix 20 rpm/20 min, and sedimentation time 12 h—and the ozonation process’ best operational conditions—pH 4.0, [Fe^2+^] = 1.0 mM, ozone flow rate 5 mg/min, air flow 1.0 L/min, agitation 350 rpm, and UV-C radiation (254 nm).

Figure 9 shows a TOC removal of 63.2 and 66.1%, respectively, after the performance of O_3_ and the combined O_3_/CFD processes (Appendix A). A TOC removal of 44.6 and 65.5%, respectively, was also observed, after the performance of CFD and the CFD/O_3_ processes. Clearly, there were not significant differences between both the combined treatments; however, after the analysis of the biodegradability, an increase, from 0.28 (raw WW) to 0.29 for O_3_/CFD and 0.40 for CFD/O_3_, was observed. Therefore, the application of CFD before the O_3_ process can be more advantageous for the WW treatment. These results were in agreement with Liu et al. [94], who observed that the performance of the CFD process, as a pre-treatment, enhanced the ozonation process in the treatment of a landfill leachate. The results were also in agreement with Bu et al. [95], who observed that the performance of the CFD process, as a pre-treatment, followed by ozonation, enhanced the UV_254_ removal efficiency and decreased the accumulation of organic matter in the treatment of wastewater. Considering the Portuguese Decree Law nº 236/98 for residual water discharge, only TSS achieved the legal value (60 mg/L), while COD (150 mg O_2_/L), BOD_5_ (40 mg O_2_/L), and total polyphenols (0.5 mg gallic acid/L), failed to reach the legal thresholds. Nonetheless, the observed improvement in biodegradability showed that the WW can be sent to a biological reactor for further degradation of organic matter.

A factor that possibly explains the high organic carbon removal that was observed after the combined O_3_/CFD and CFD/O_3_ processes, is the reduction in turbidity, TSS, and total polyphenols, which would otherwise prevent light from penetrating into the water and triggering the photo-Fenton reaction [11]. In Figure 10, the application of O_3_, O_3_/CFD, CFD, and CFD/O_3_, achieved a turbidity removal of 65.5, 99.9, 98.3, and 99.3%, respectively, and a TSS removal of 66.8, 98.3, 97.6, and 98.3%, respectively. In addition to turbidity and TSS, a total polyphenol removal of 95.1, 95.9, 81.2, and 99.3%, respectively, was also observed. These results were in agreement with Lucas et al. [23], who observed that the ozonation process could be beneficial for the removal of polyphenols (toxicity) from the winery wastewater, thus contributing to the safety of public and environmental health.

### 3.4. Effect of the Treatments in Phytotoxicity of Different Plants

In previous sections, was described how the combination of the CFD and ozonation processes were beneficial for the reduction in the organic carbon present in the WW. However, considering that the application of oenological coagulants in combination with the ozonation process, has never been implemented in wastewater treatment, its effects in vegetables are still unknown. Therefore, in this section, we evaluated the effects of the different treatments in the phytotoxicity of different species of vegetables. A series of tests were performed on the germination of seeds, in order to evaluate the phytotoxicity of the treatments, similarly to several authors [96,97,98,99], in two Dicotyledonae species (lettuce and cucumber) and in two Monocotyledonae species (corn and onion). In Figure 11, it was observed that WW had a phytotoxic effect on cucumber and lettuce seeds (GI = 51 and 0%, respectively). The performance of the treatments O_3_, O_3_/CFD, CFD, and CFD/O_3_ increased the germination index (GI) of the cucumber seeds, to 113, 109, 71, and 117% (Appendix A), respectively. The germination index of the lettuce seeds increased to 298, 88, and 249%, respectively, for O_3_, O_3_/CFD, and CFD/O_3_. These are excellent values that reveal the goodness of the studied treatments and the reduction in phytotoxicity.

As previously observed, some of the problems that are generated by this type of wastewater are the intense color and high turbidity, which absorbs the radiation and decreases the treatment efficiency. The intense color that is present in winery wastewaters appears mainly due to the presence of total phenols, non-flavonoids, flavonoids, total anthocyanins, colored anthocyanins, total pigments, and total tannins, present in the wines [100]. Considering that there is insufficient information regarding the impact of the CFD and ozonation processes in the phenolic composition of the WW, in this section, the efficiency of these treatments in the removal of total phenols, non-flavonoids, flavonoids, total anthocyanins, colored anthocyanins, total pigments, and total tannins, was evaluated (Figure 12). The performance of the treatments O_3_, O_3_/CFD, CFD, and CFD/O_3_ achieved a total phenol removal of 0.0, 5.1, 1.9, and 3.2%, respectively, a non-flavonoid removal of 0.0, 0.0, 2.6, and 2.6%, respectively, and a flavonoid removal of 0.0, 14.6, 0.0, and 7.3% (Appendix A), respectively, which are responsible for the yellow color in wines [38]. In Figure 12, a high decrease in the total anthocyanins (50.0, 50.0, 25.0, and 100%, respectively) and colored anthocyanins (28.6, 10.0, 42.9, and 100%, respectively) was observed, which are linked to the red color of wines [101]. The reduction in these phenolic compounds had an effect in the removal of color from the wastewater, which was evaluated by a CIELab analysis (Appendix A). The combined treatments O_3_/CFD and CFD/O_3_ had negative values for ∆a* and ∆b* (−1.90 and −3.29), which indicated a reduction in the red and yellow color. The luminosity (L^*^) increased from 0.0% (raw WW) to 99.7 and 100%, respectively, which meant that the phenolic compounds were directly linked to the dark-yellow color of the wastewater. The color removal, given by the Euclidean distance, was 99.74 and 100, after the combined treatments of O_3_/CFD and CFD/O_3_, which meant that color removal was perceptible by the human eye, which is in accordance with Spagna et al. [41].

## 4. Conclusions

In this work, a WW was treated by two different processes, a CFD process, which employed activated sodium bentonite mixed with potassium caseinate, and the advanced oxidation process O_3_/Fe^2+^/UV-C. The combination of both treatment processes in a CFD/O_3_/Fe^2+^/UV-C system, proved to be a feasible method for the treatment of WW, and the main conclusions are summarized as follows:

The performance of a CFD process, by application of an SLD statistical design, allows a high removal of turbidity, TSS, TOC, and COD (98.3, 97.6, 44.6, and 48.0%);The application of an ozonation process, under the best operational conditions—pH = 4.0, [Fe^2+^] = 1.0 mM, ozone flow rate 5 mg/min, air flow 1.0 L/min, agitation 350 rpm, time 600 min, and radiation UV-C mercury lamp (254 nm)—achieves a TOC removal of 63.2%;The O_3_/1.0 mM Fe^2+^/UV-C system is concluded to be very efficient in terms of energy consumption, with an $E_{\mathrm{EO}}$ = 1843 kWh m^−3^ order^−1^;The combined processes O_3_/CFD and CFD/O_3_ achieved high TOC removal (66.1 and 65.5%, respectively). It is also concluded that the performance of the CFD/O_3_ process achieves higher biodegradability (0.40);It is concluded that the combined processes O_3_/CFD and CFD/O_3_ have lower phytotoxicity effects in the germination of plant seeds;The combined process O_3_/CFD and CFD/O_3_ have the capacity to completely decolor the WW (L* = 100%), through the high removal of phenolic compounds;The combined process O_3_/CFD and CFD/O_3_ is concluded to decrease the risk of public and environmental health problems.

These results showed that the combination of both treatments was essential to achieve the high degradation of organic matter from the WW. In the future, based on these results, new approaches can be explored, such as optimization of the hydraulic retention time and microbial elimination.

## Figures and Tables

**Figure 1 ijerph-18-08882-f001:**
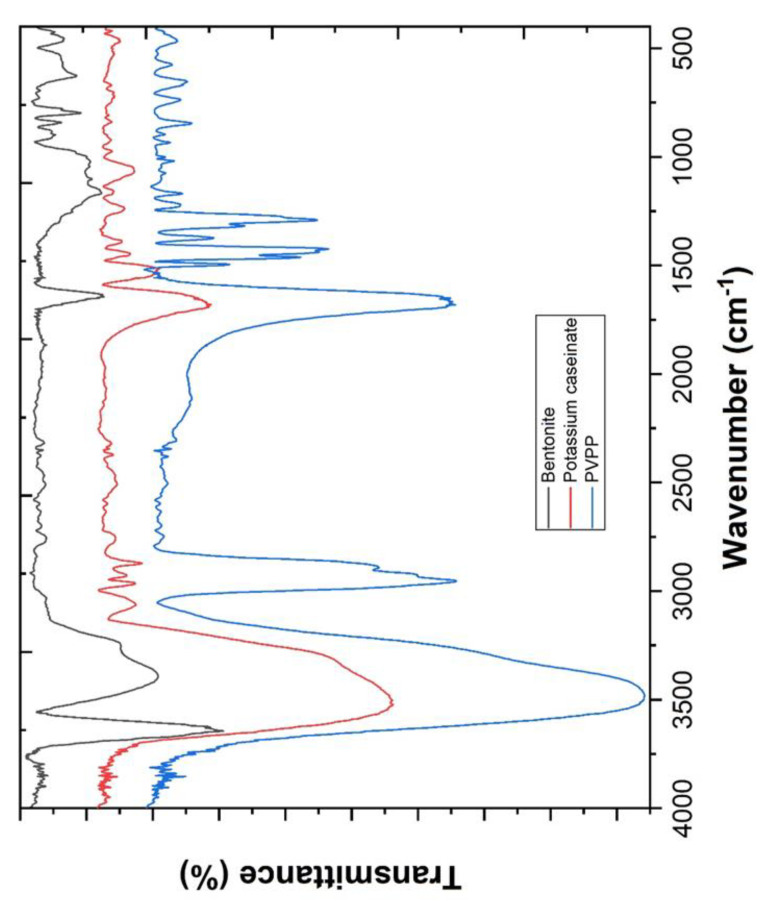
FTIR spectra of bentonite, potassium caseinate and PVPP.

**Figure 2 ijerph-18-08882-f002:**
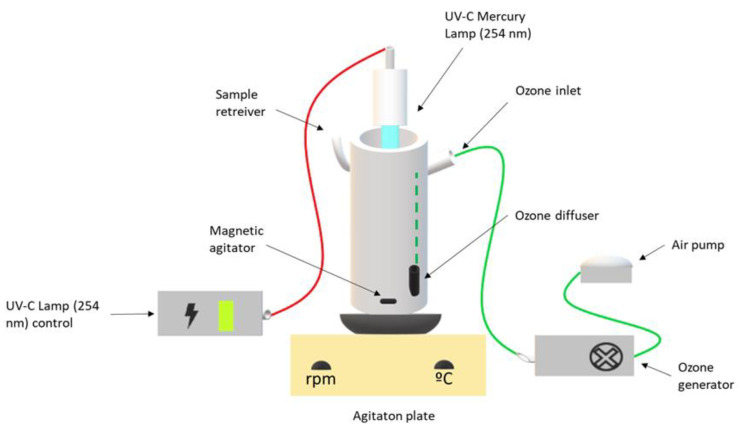
Schematic representation of ozonation and UV reactor.

**Figure 3 ijerph-18-08882-f003:**
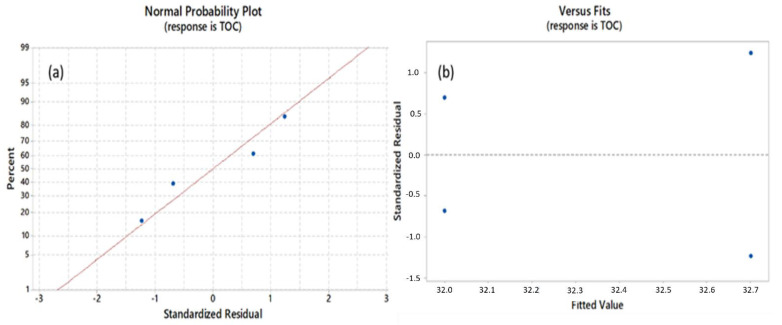
(**a**) Normal probability plot, (**b**) residual versus the fitted value of the data after CFD process, with TOC as response.

**Figure 4 ijerph-18-08882-f004:**
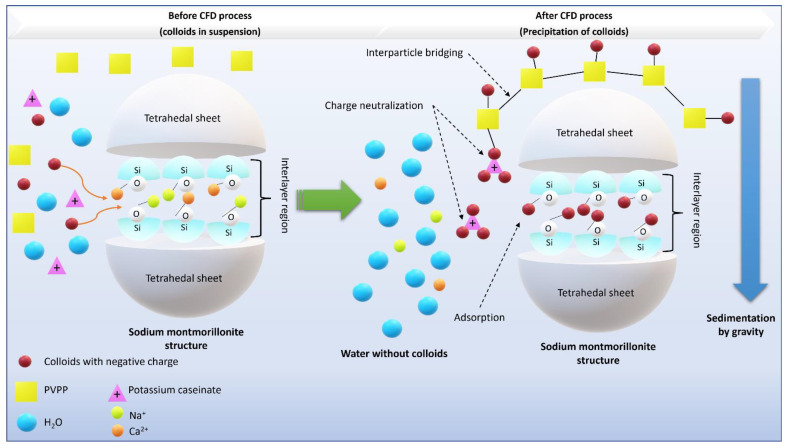
Mechanism of colloid removal by activated sodium bentonite, potassium caseinate and PVPP.

**Figure 5 ijerph-18-08882-f005:**
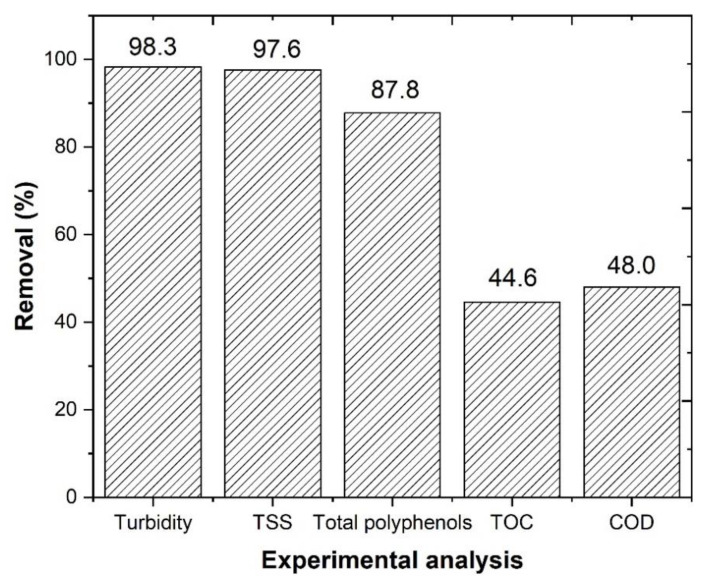
Overall results after coagulation–flocculation–decantation process (CFD). Operational conditions: 0.48 g/L potassium caseinate, 0.52 g/L bentonite, pH 4.0, temperature 298 K, rapid mix 150 rpm/3 min, slow mix 20 rpm/20 min an 12 h of sedimentation time.

**Figure 6 ijerph-18-08882-f006:**
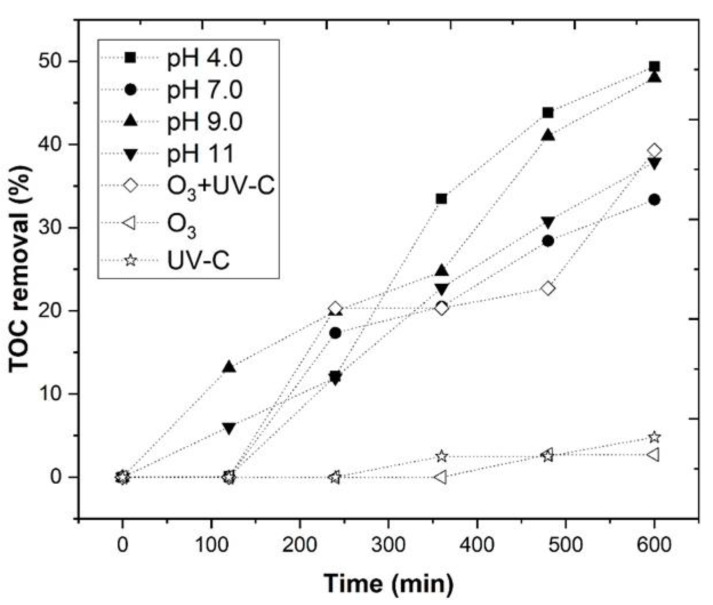
Evaluation of TOC removal through the O_3_/Fe^2+^/UV-C process at different pH values (4.0–11.0). The following ozonation experimental conditions: [Fe^2+^] = 1.0 mM, ozone flow rate 5 mg/min, air flow 1.0 L/min, agitation 350 rpm, time 600 min, radiation UV-C mercury lamp (254 nm). Blank experiments (O_3_/UV-C, O_3_ and UV-C—pH 4.0) are also shown as a reference.

**Figure 7 ijerph-18-08882-f007:**
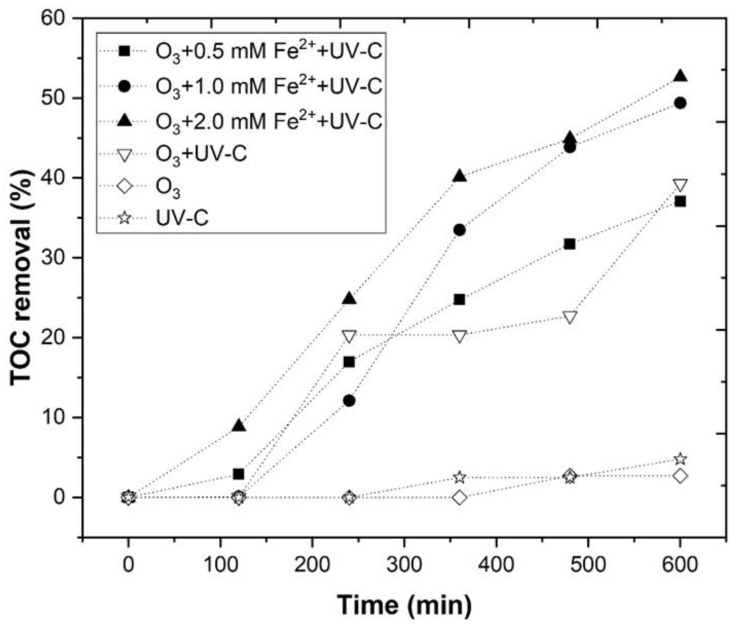
Evaluation of TOC removal through the O_3_/Fe^2+^/UV-C process at different Fe^2+^ concentrations (0.5–2.0 mM). Ozonation experimental conditions were as follows: pH = 4.0, ozone flow rate 5 mg/min, air flow 1.0 L/min, agitation 350 rpm, time 600 min, radiation UV-C mercury lamp (254 nm). Blank experiments (O_3_/UV-C, O_3_, and UV-C—pH 4.0) are also shown as a reference.

**Figure 8 ijerph-18-08882-f008:**
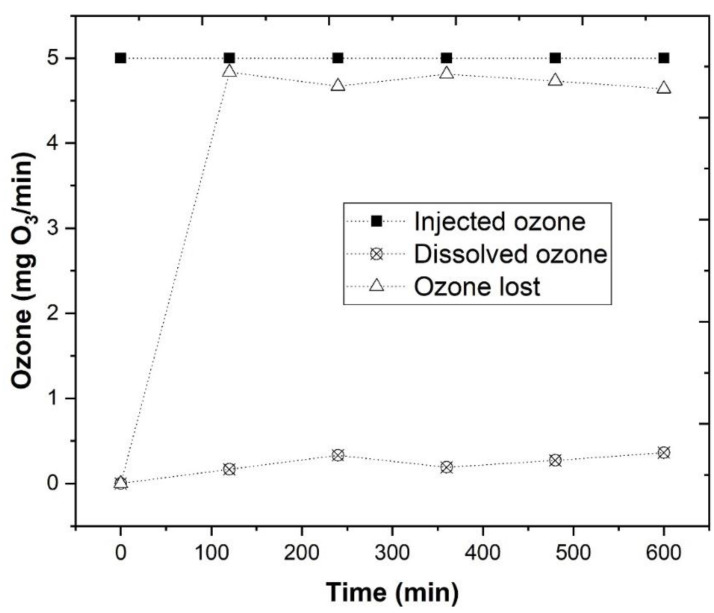
Evaluation of the O_3_ concentration dissolved within the WW, by application of the indigo ozone reagent. O_3_/Fe^2+^/UV-C experimental conditions were as follows: pH = 4.0, [Fe^2+^] = 1.0 mM, ozone flow rate 5 mg/min, air flow 1.0 L/min, agitation 350 rpm, time 600 min, radiation UV-C mercury lamp (254 nm).

**Figure 9 ijerph-18-08882-f009:**
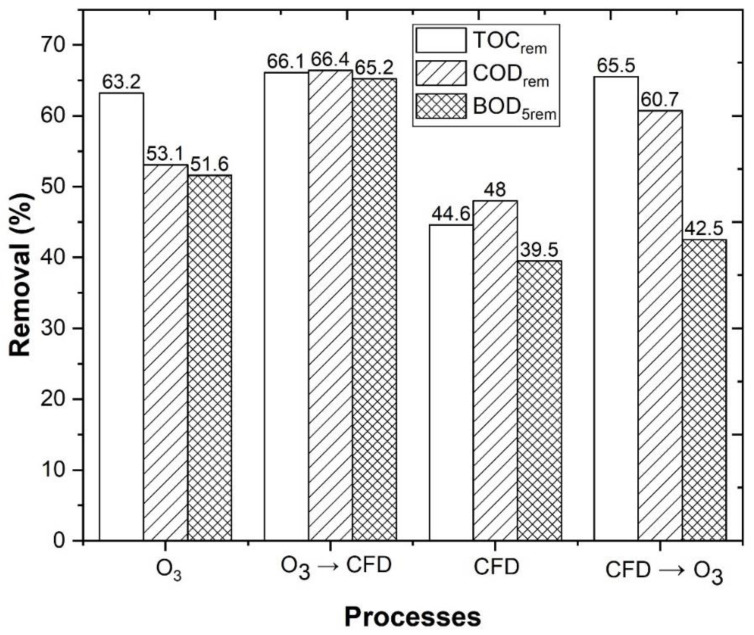
TOC, COD and BOD_5_ removal. CFD operational conditions, as follows: 0.48 g/L potassium caseinate, 0.52 g/L bentonite, pH 4.0, temperature 298 K, rapid mix 150 rpm/3 min, slow mix 20 rpm/20 min, sedimentation time 12 h. O_3_/Fe^2+^/UV-C experimental conditions, as follows: pH = 4.0, [Fe^2+^] = 1.0 mM, ozone flow rate 5 mg/min, air flow 1.0 L/min, agitation 350 rpm, time 600 min, radiation UV-C mercury lamp (254 nm).

**Figure 10 ijerph-18-08882-f010:**
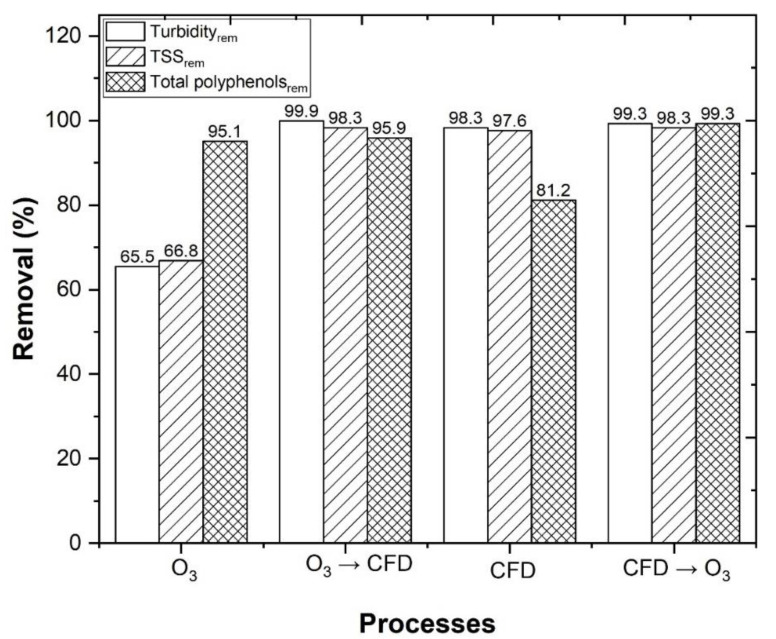
Turbidity, TSS and total polyphenol removal. The following CFD operational conditions: 0.48 g/L potassium caseinate, 0.52 g/L bentonite, pH 4.0, temperature 298 K, rapid mix 150 rpm/3 min, slow mix 20 rpm/20 min, sedimentation time 12 h. O_3_/Fe^2+^/UV-C experimental conditions, as follows: pH = 4.0, [Fe^2+^] = 1.0 mM, ozone flow rate 5 mg/min, air flow 1.0 L/min, agitation 350 rpm, time 600 min, radiation UV-C mercury lamp (254 nm).

**Figure 11 ijerph-18-08882-f011:**
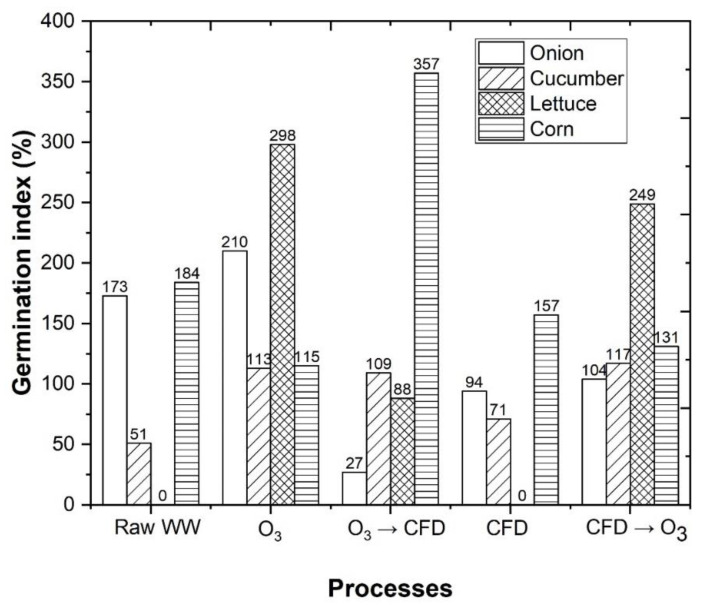
Analysis of germination index (GI) regarding the germination of onion, cucumber, lettuce and corn after ozonation (O_3_), coagulation–flocculation–decantation (CFD) and combined O_3_ → CFD and CFD → O_3_ treatments. The following CFD operational conditions were used: 0.48 g/L potassium caseinate, 0.52 g/L bentonite, pH 4.0, temperature 298 K, rapid mix 150 rpm/3 min, slow mix 20 rpm/20 min, sedimentation time 12 h. The following O_3_/Fe^2+^/UV-C experimental conditions were used: pH = 4.0, [Fe^2+^] = 1.0 mM, ozone flow rate 5 mg/min, air flow 1.0 L/min, agitation 350 rpm, time 600 min, radiation UV-C mercury lamp (254 nm). IG ≤ 50% (high concentration of phytotoxic substances), 80% < IG > 50% (moderated presence of phytotoxic substances), IG ≥ 80% (there are no phytotoxic substances, or they exist in very small dosages).

**Figure 12 ijerph-18-08882-f012:**
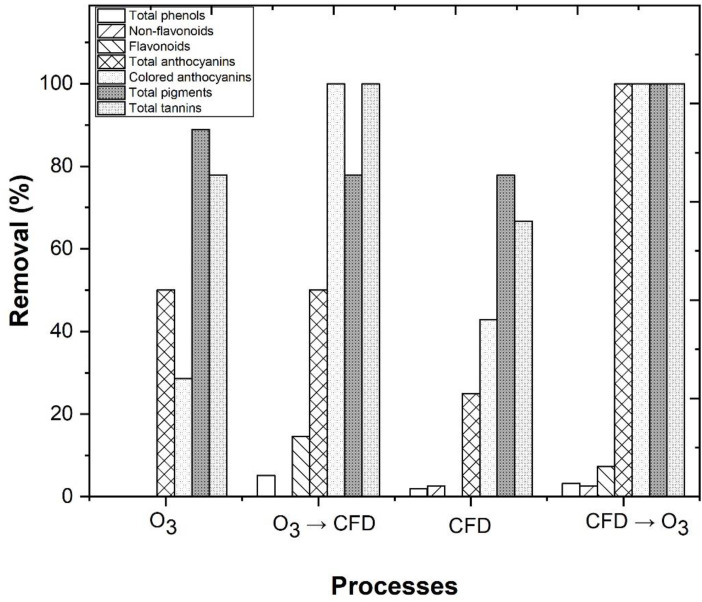
Analysis of total phenols, non-flavonoids, flavonoids, total anthocyanins, colored anthocyanins, total pigments and total tannins removal. The following CFD operational conditions: 0.48 g/L potassium caseinate, 0.52 g/L bentonite, pH 4.0, temperature 298 K, rapid mix 150 rpm/3 min, slow mix 20 rpm/20 min, sedimentation time 12 h. The following O_3_/Fe^2+^/UV-C experimental conditions: pH = 4.0, [Fe^2+^] = 1.0 mM, ozone flow rate 5 mg/min, air flow 1.0 L/min, agitation 350 rpm, time 600 min and a UV-C mercury lamp (254 nm).

**Table 1 ijerph-18-08882-t001:** Winery wastewater characteristics.

Parameter	Value
pH	4.0
Conductivity (µS/cm)	475
Turbidity (NTU)	1040
Total suspended solids (mg/L)	2430
Chemical oxygen demand (mg O_2_/L)	9432
Biochemical oxygen demand (mg O_2_/L)	2611
Total organic carbon (mg C/L)	1962
Total polyphenols (mg gallic acid/L)	123
Ferrous iron (mg Fe/L)	0.05
Biodegradability index—BOD_5_/COD	0.28

**Table 2 ijerph-18-08882-t002:** Formulas for phenolic composition and chromatic (CIELab) determination.

Formulas	Parameters	References
Color intensity (CI)	A_420_—absorbance at 420 nm	OIV, [35]
CI = A_420_ + A_520_ + A_620_	A_520_—absorbance at 520 nm	
Hue	A_620_—absorbance at 620 nm	OIV, [35]
$\mathrm{Hue}=\frac{A_{420}}{A_{520}}$	A_280_—absorbance at 280 nm	
Total polyphenol index (TPI)	DF—dilution factor	Curvelo-Garcia, [36]
TPI = A_280_*DF		
Total phenols		Kramling and Singleton, [37]
$\mathrm{Total}\;\mathrm{phenols}\;(\mathrm{mg}\;\mathrm{gallic}\;\mathrm{acid}/L)=\frac{A_{280}^{\mathrm{IPT}}+0.0243}{0.0326}*\mathrm{DF}$		
Non-flavonoids		Kramling and Singleton, [37]
$\mathrm{Non}-\mathrm{flavonoids}\;(\mathrm{mg}\;\mathrm{gallic}\;\mathrm{acid}/L)=\frac{A_{280}^{\mathrm{NF}}+0.0243}{0.0326}*\mathrm{DF}$		
Flavonoids		Kramling and Singleton, [37]
Flavonoids (mg gallic acid/L) = total phenols–non-flavonoids		
Total anthocyanins (C)		Ribéreau-Gayon et al. [38]
C (mg/L) = 875*(A_1_–A_2_)	A_1_/A_2_—absorbance at 520 nm	
Colored anthocyanins (CA)		Somers and Evans [39]
$\mathrm{CA}\;(\mathrm{mg}/L)=(A_{520}^{\mathrm{no}\;\mathrm{bisulfite}}$$*10)–(A_{520}^{\mathrm{bisulfite}}$*10)		
Total pigments (TP)		Somers and Evans [39]
$\mathrm{TP}\;(\mathrm{mg}/L)=A_{520}^{\mathrm{HCl}}$*10		
Polymeric pigments (PP)		Somers and Evans [39]
$\mathrm{PP}\;(\mathrm{mg}/L)=A_{520}^{\mathrm{HCl}}$*10		
Total tannins (L.A.)		Ribéreau-Gayon and Stonestreet [40]
L.A. (g/L) = 19.33*(D_2_–D_1_)	D_1_/D_2_—absorbance at 520 nm	
CIELab	*L*—lightness	Schanda [42]
∆*L* = $L_{1}$ − $L_{0}$	*a*—redness	
∆*a* = $a_{1}$ − $a_{0}$	*b*—yellowness	
∆*b* = $b_{1}$ − $b_{0}$		
∆*E*_ab_ = [($\Delta {L)}^{2}$ + ($\Delta {a)}^{2}$ + ($\Delta {b)}^{2}$]		

**Table 3 ijerph-18-08882-t003:** Chemical composition of activated sodium bentonite by EDS/EDAX.

Element	Mass Concentration (wt %)
Si	69.49
Al	17.67
Fe	2.95
Mg	2.73
Ca	2.00
Na	2.76
K	1.37
S	1.03

**Table 4 ijerph-18-08882-t004:** Specific surface areas and pore characteristics of coagulants (n.q.—not quantifiable).

Coagulants	S_BET_ (m^2^/g)	V_total pore_ (cm^3^/g)	Particle Size (nm)
Activated sodium bentonite	8.8	0.045	4.0
Potassium caseinate	1.0	n.q.	n.q.
PVPP	n.q.	n.q.	n.q.

**Table 5 ijerph-18-08882-t005:** Design matrix with the experimental and predicted removal percentages of turbidity, TSS, COD and TOC. Experimental conditions: pH 4, temperature 298 K, rapid mix 150 rpm/3 min, slow mix 20 rpm/20 min, sedimentation time 12 h. X_1_—potassium caseinate, X_2_—bentonite, X_3_—PVPP.

Experiments	Samples			Y_1_: Turbidity		Y_2_: TSS		Y_3_: COD		Y_4_: TOC	
	X_1_	X_2_	X_3_	Observed	Predicted	Observed	Predicted	Observed	Predicted	Observed	Predicted
CFD1	0.00	1.00	0.00	99.6	99.6	98.3	98.3	54.3	54.4	28.4	28.4
CFD2	0.67	0.17	0.17	99.5	99.5	98.0	98.0	52.9	52.5	32.5	32.4
CFD3	1.00	0.00	0.00	99.5	99.5	97.9	97.9	48.5	48.6	31.6	31.6
CFD4	0.17	0.17	0.67	99.3	99.3	97.9	97.9	52.1	51.7	29.6	29.5
CFD5	0.00	0.00	1.00	98.9	98.9	97.5	97.5	56.2	56.3	37.0	37.0
CFD6	0.33	0.33	0.33	99.5	99.5	98.0	98.1	50.7	51.7	31.6	31.9
CFD7	0.17	0.67	0.17	99.7	99.7	98.3	98.3	52.5	52.1	34.4	34.3

**Table 6 ijerph-18-08882-t006:** Evaluation of first-order kinetic rate (k), half-life ($t_{1/2}$) and electric energy per order ($E_{\mathrm{EO}}$) through the O_3_/Fe^2+^/UV-C process at different Fe^2+^ concentrations (0.5–2.0 mM). *P*_Ozonator_ = 0.025 kW, *P*_UV-C_ = 0.015 kW, *t* = 10 h, V = 0.5 L.

[Fe^2+^]	k	$t_{1/2}$	$E_{\mathrm{EO}}$
mM	(min^−1^)	(min)	(kWh∙m^−3^∙order^−1^)
UV-C	7.17 × 10^−4^	966.5	1720
O_3_/UV-C	1.31 × 10^−3^	529.0	2153
O_3_	6.86 × 10^−4^	1009.9	2996
O_3_/0.5 mM Fe^2+^/UV-C	1.34 × 10^−3^	517.2	2065
O_3_/1.0 mM Fe^2+^/UV-C	1.67 × 10^−3^	414.9	1843
O_3_/2.0 mM Fe^2+^/UV-C	1.72 × 10^−3^	402.9	1677

**Table 7 ijerph-18-08882-t007:** Reported values of electric energy per order ($E_{\mathrm{EO}}$ ) in ozonation processes.

Wastewater Type	AOP Process	Observations	$E_{\mathrm{EO}}\;(\mathrm{kWh}\cdot m^{-3}\cdot {\mathrm{order}}^{-1})$	References
Winery wastewater	UV-C (254 nm)/PMS/Co(II)	[PMS] = 2.5 mM [Co(II)] = 1.0 mM *t* = 90 min ${\mathrm{TOC}}_{i}$ = 143 mg C/L	173	[19]
Active pharmaceutical ingredients (APIs)	UV-C/O_3_	Gas flow = 3.2 L/min *t* = 30 min ${\mathrm{TOC}}_{i}$ = 21.5 mg C/L	1.50	[91]
Real textile wastewater	Direct ozonation	Gas flow = 1.4 L/min *t* = 9 min ${\mathrm{TOC}}_{i}$ = 169 mg C/L	2.43	[92]
Organic pollutants	UV-C (254 nm)/O_3_	Gas flow = 0.4 L/min *t* = 15 min TOC_i_ = 79 mg C/L	29.10	[93]
	UV-C (254 nm)/TiO_2_/O_3_	Gas flow = 0.4 L/min *t* = 20 h TOC_i_ = 79 mg C/L	10.23	
Winery wastewater	Fe^2+^/O_3_/Fe^2+^/UV-C (254 nm)	Gas flow = 1.0 L/min [Fe^2+^] = 1.0 mM *t* = 10 h TOC_i_ = 1962 mg C/L	1843	Present results

## Data Availability

Not applicable.

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
