# Peer review of "Combination of Coagulation–Flocculation–Decantation and Ozonation Processes for Winery Wastewater Treatment"

_ijerph, 2021, doi:10.3390/ijerph18168882_

Round 1

Reviewer 1 Report

The manuscript named “Combination of coagulation-flocculation-decantation and ozonation processes for winery wastewater treatment” reports a combination of several methods focused to the treatment of winery wastewater (WW).

A complete WW treatment (including coagulation, flocculation, decantation, ozonation) was systematically applied following a model optimization. The phytotoxicity of plant seeds germination and the changes in phenolic composition after treatment were as well included.

In general, overall quality of the data and interpretations of them are good. The manuscript contains interesting results, and it may be published in the International Journal of Environmental Research and Public Health.

Author Response

Dear Reviewer 1, we are very grateful for your positive comments and for the thorough and detailed review of our manuscript. Firstly, we would like to say that winery wastewater was selected for this study due to the high number of wineries that exist in Douro region (north of Portugal), from large wineries producing millions of hectoliters to smaller wineries producing a few thousands of liters. From a more global point of view, the wine effluents problem affects the main wine producing countries in the world. The winery wastewater has a very dark color, with high content of recalcitrant compounds and odors that can cause large problems if released into the environment without proper treatment.

This work intended to (1) decrease the high organic content from this type of wastewater, (2) increase its biodegradability, (3) explore non-toxic organic coagulants for coagulation-flocculation-decantation process and (4) study the performance of ozonation in the reduction of organic carbon from the winery wastewater.

This work presented a novel idea, which was the use of a mixture of potassium caseinate and activated sodium bentonite for coagulation-flocculation-decantation process (CFD process). These products are used in wine treatment, they are organic and not toxic. In addition to CFD process, it was studied the application of ozonation/ferrous iron/UV-C system for the first time, for the treatment of winery wastewater. The results showed a high degradation of organic carbon within 10 h of reaction. The combination of both processes increased organic carbon and total polyphenols degradation and biodegradability. The inclusion of germination index test showed that these treatments pose little threat to the environment.

Reviewer 2 Report

This manuscript reports the treatment of a WW by coagulation-flocculation-decantation and ozonation processes. The logic and significance of the work sound well. The experimental setup and results are properly provided. However, further improvements should be done before acceptance as follow:

  • Although the authors obtained good results concerning turbidity, TSS and TOC removal, no removal mechanism was provided for any.

  • The authors used polyvinylpyrrolidone (PVP not PVPP) during WW treatment without declaration what it's actual role.

  • The author stated on page 9 line 271 that “Activated sodium bentonite has an isoelectric point of 7, becoming electronegative at pH 3.6 – 4.0, which made it an ideal choice for WW treatment”. It seems that this point is miswritten as bentonite should have a net negative charge as a structural property, not an amphoteric particle. During the treatment process, bentonitic clays delaminate to very small nanosheets. These nanosheets are believed to be the nuclei for the flocculation process. I wonder what the actual pH of the used bentonite suspension is. How bentonite can assist the removal of contaminants from WW in this study? What is the mechanism?

  • Although the authors get a TOC removal of 59.0, 63.2 and 66.7%, respectively, for 0.5, 1.0 and 2.0 mM Fe2+. They decided to use 1.0 mM Fe2+ due to the high cost associated with the application of 2.0 mM Fe2+!!! I never think that the usage of FeSO4 is expensive. Instead, authors should reduce the irradiation time of UV light to 240 min as it yields almost similar results as for 600 min, to reduce the operation costs. In addition, I suggested continuing to increase the concentration of Fe2+ up to 0.1 M (5 mM, 7mM and 10 mM) and checking the efficacy of the process to remove TOC.

  • The author reported that “The performance of treatments O3, O3/CFD, CFD and CFD/O3 achieved a total phenols removal of 0.0, 5.1, 1.9 and 3.2%, respectively, non-flavonoids removal of 0.0, 0.0, 2.6 and 2.6%, respectively and flavonoids removal of 0.0, 14.6, 0.0 and 7.3%”. It seems that the adopted method failed to treat these compounds; however, the addition of H2O2 can make a difference. I strongly suggest considering the addition of H2O2 in an optimized manner.

  • English typos are found eg.:
  • ferric sulphate heptahydrated (FeSO47H2O) by Panreac. Should be ferrous sulphate heptahydrate.
  • Blanc should be blank.
  • Please revise the language thoroughly.

Author Response

This manuscript reports the treatment of a WW by coagulation-flocculation-decantation and ozonation processes. The logic and significance of the work sound well. The experimental setup and results are properly provided. However, further improvements should be done before acceptance as follow:

  1. Although the authors obtained good results concerning turbidity, TSS and TOC removal, no removal mechanism was provided for any.

Thank you, Reviewer 2, we thank you for this valuable suggestion. Indeed, no formula had been presented for degradation of turbidity, TSS and TOC. In page 3, line 135 – 138, it is presented Equation 2, which shows the mechanism for turbidity, TSS and TOC removal.

  1. The authors used polyvinylpyrrolidone (PVP not PVPP) during WW treatment without declaration what it's actual role.

Reviewer 2, thank you for the valuable insight in this comment. Indeed, it is polyvinylpolypyrrolidone (PVPP) and not polyvinylpyrrolidone (PVP). This error was corrected in the text (page 2, line 57, page 3, line 100). The PVPP was selected due to its ability to remove polyphenols from the wastewater. In this work it was optimized the mixture of 3 compounds (potassium caseinate, activated sodium bentonite and PVPP) by the statistical program Simplex Lattice Design. In page 10, line 318 – 321 it is showed the mechanism of PVPP. Despite the positive effect of PVPP in this work, the statistical program showed that PVPP could be removed from the mixture, thus decreasing the costs associated, as it was observed in Figure S.1.

  1. The author stated on page 9 line 271 that “Activated sodium bentonite has an isoelectric point of 7, becoming electronegative at pH 3.6 – 4.0, which made it an ideal choice for WW treatment”. It seems that this point is miswritten as bentonite should have a net negative charge as a structural property, not an amphoteric particle. During the treatment process, bentonitic clays delaminate to very small nanosheets. These nanosheets are believed to be the nuclei for the flocculation process. I wonder what the actual pH of the used bentonite suspension is. How bentonite can assist the removal of contaminants from WW in this study? What is the mechanism?

Reviewer 2, the authors greatly appreciate your valuable contribute, which helps to improve the work’s quality. The activated sodium bentonite was added directly without dilution and the wastewater pH was controlled to avoid changes in pH. In page 10, line 310 – 318, it is showed that activated bentonite removes the organic matter from the wastewater by an adsorption mechanism. Due to the difference of charges at pH 4.0, bentonite’s efficiency is increased as observed by other authors.

  1. Although the authors get a TOC removal of 59.0, 63.2 and 66.7%, respectively, for 0.5, 1.0 and 2.0 mM Fe2+. They decided to use 1.0 mM Fe2+due to the high cost associated with the application of 2.0 mM Fe2+!!! I never think that the usage of FeSO4is expensive. Instead, authors should reduce the irradiation time of UV light to 240 min as it yields almost similar results as for 600 min, to reduce the operation costs. In addition, I suggested continuing to increase the concentration of Fe2+ up to 0.1 M (5 mM, 7mM and 10 mM) and checking the efficacy of the process to remove TOC.

We thank you for your valuable suggestion. Indeed, with the increase of Fe2+ concentration there was an increase of TOC removal. However, in page 13, line 419 – 420, it was observed an Fe2+ leaching of 1.84, 12.60 and 41.92 mg Fe/L, respectively, for 0.5, 1.0 and 2.0 mg/L. Considering that Portuguese Law Decree nº 236/98 for residual water discharge only allows a maximum of 2.0 mg Fe/L, the application of 2.0 mM Fe2+ already has a high concentration of Fe leached from this treatment. The application of higher Fe2+ concentrations will only increase the Fe2+ in solution and create further cost for iron removal after treatments.

The authors will take Reviewer 2 suggestion in consideration and try to apply higher Fe2+ concentrations in future works.

  1. The author reported that “The performance of treatments O3, O3/CFD, CFD and CFD/O3achieved a total phenols removal of 0.0, 5.1, 1.9 and 3.2%, respectively, non-flavonoids removal of 0.0, 0.0, 2.6 and 2.6%, respectively and flavonoids removal of 0.0, 14.6, 0.0 and 7.3%”. It seems that the adopted method failed to treat these compounds; however, the addition of H2O2can make a difference. I strongly suggest considering the addition of H2O2 in an optimized manner.

Reviewer 2, thank you for your valuable suggestion. Indeed, several authors reported the combined use of ozone with H2O2 as a very efficient combination for organic removal from the wastewater. However, H2O2 is very expensive, and Equation 10 and 11 showed that application of ozone under UV radiation can generate H2O2. Therefore, one of this works objectives was the use of ozone as the oxidant agent, because it is cheaper to produce ozone from an air supplier. The authors will take Reviewer 2 advice in consideration and apply in future works ozone in combination with H2O2.

  1. English typos are found e.g.:

ferric sulphate heptahydrated (FeSO47H2O) by Panreac. Should be ferrous sulphate heptahydrate.

Blanc should be blank.

These errors were corrected (page 3, line 99 – 103). Al “Blancs” were replaced by “Blank”.

  1. Please revise the language thoroughly.

Thank you, Reviewer 2, all language was reviewed thoroughly.

Reviewer 3 Report

Wastewater treatment, in general, is of significant concern in the world, as if not properly treated can pose a significant threat to human health and the environment in general. The concern is also evident in the field of winery production as one of the growing industrial sectors producing highly polluted wastewater due to the nature of the production activities. In that matter, the current study is relevant and has a potential interest to the readers of the journal.

In this study, the authors tried to investigate the performance of an integrated wastewater treatment system; where winery wastewater effluents were remediated by combining the coagulation-flocculation-decantation processes with the ozonation process. However, the CFD process is not new in the field of wastewater treatment, and even the combination with ozonation is generally not new. The authors could have concentrated their novelty on the type of wastewater to be treated by such a treatment approach, considering that the same treatment approach can perform differently when subjected to wastewater with different characteristics.

More specific comments are provided below;

  1. The abstract part needs to be revised and modified to present only the most relevant goal and idea. It is essential to highlight the material merits and work novelty. In addition, there are some details about the process that should be reduced.
  2. Also, I wouldn’t recommend the use of abbreviations in the abstract
  3. In the introduction section, the authors should also highlight the advantage of ozone in the oxidation process in front of its competitors. For example, what could have been a disadvantage of using chlorine in the oxidation process in place of ozone?
  4. The type of sampling and the number of samples used in the statistical analysis should be highlighted in the manuscript.
  5. In table 1, the TSS has been listed but the analytical process of total suspended solids has not been described in the text.
  6. The experimental setup is not clear, as to how much wastewater goes to the system, an estimate of how much gets out as purified water, and an estimate of sludge. The hydraulic retention time for each sub-unit within the integrated system should be presented. Some information like power consumption of each sub-unit or the entire treatment plant could add a piece of more useful information.
  7. The role of each sub-unit in the integrated treatment approach is not clear
  8. The authors should also include latitude and longitudes of the case study, with a clear reference (for example: certain km from the capital city or international airport), as well as briefly present the nature of activities at the industry
  9. All analytical methods including test kits and reagents with the manufacturer’s name, city and country should be well presented.
  10. It could have been more interesting if the authors also checked the performance of the treatment approach for microbial elimination.
  11. Could also be interesting if the authors presented a schematic diagram of the treatment plant
  12. All statistical methods used in the study with their respective tools should be highlighted in the materials and methods section
  13. With the fact that the experimental setup has not been well presented, it becomes difficult to properly follow the results and discussion section.
  14. In most cases, you will find that achieving relatively high removal efficiencies is fine, but the removal efficiencies lead to effluent quality to be used for what purpose becomes the question. Therefore, a comparison with some recommended water quality standards or guidelines could provide a more useful piece of information.
  15. The quality of the figures should also be improved.
  16. I would suggest designing the conclusion section as one paragraph based on the objective, approach, and the results’ merits, work novelty, and future applicability.
  17. In general, the paper should be thoroughly checked in terms of writing style, e.g., L24: “WW treatment can be an efficient treated”. This statement is not grammatically correct.

Author Response

"Wastewater treatment, in general, is of significant concern in the world, as if not properly treated can pose a significant threat to human health and the environment in general. The concern is also evident in the field of winery production as one of the growing industrial sectors producing highly polluted wastewater due to the nature of the production activities. In that matter, the current study is relevant and has a potential interest to the readers of the journal.

In this study, the authors tried to investigate the performance of an integrated wastewater treatment system; where winery wastewater effluents were remediated by combining the coagulation-flocculation-decantation processes with the ozonation process. However, the CFD process is not new in the field of wastewater treatment, and even the combination with ozonation is generally not new. The authors could have concentrated their novelty on the type of wastewater to be treated by such a treatment approach, considering that the same treatment approach can perform differently when subjected to wastewater with different characteristics."

Reviewer 3, thank you for your observations, they were very helpful to improve the quality of this work. First, the authors would like to say that reviewer 3 is correct, the application of coagulation-flocculation-decantation (CFD) and ozonation processes are not knew in wastewater treatment. However, the application of potassium caseinate and activated sodium bentonite in CFD process for the treatment of winery wastewater are knew and they have potential to replace metallic coagulants such as aluminum and iron metal salts. However, their effects in the removal of turbidity, total suspended solids, total organic carbon and total polyphenols from winery wastewater are unknown, which justifies this work.

In addition, in this work it was studied for the first time the application of an ozone/ferrous iron/UV-C system in the treatment of winery wastewater. A statistical study performed in web of science showed that from 2005 to 2021, there were only 15 publications regarding the application of ozonation process in winery wastewater treatment. Due to the low number of publications, there are a lot of variables that remain unanswered, which justifies the performance of this work. In this work, it was studied the combination of ozone with ferrous iron under UV-C radiation.

Reviewer 3 also wrote that “The authors could have concentrated their novelty on the type of wastewater to be treated by such a treatment approach”. In page 2, line 56 – 59 and page 2, line 77 – 80 it is indicated that these treatments were never performed before in winery wastewater treatment. The authors believe that the novelty of this article is not only in the treatments studied, but also in the type of wastewater treated and this was well demonstrated in the text.

More specific comments are provided below;

  1. The abstract part needs to be revised and modified to present only the most relevant goal and idea. It is essential to highlight the material merits and work novelty. In addition, there are some details about the process that should be reduced.

Reviewer 3, thank you for your recommendation, the authors appreciate your suggestion. The abstract was rewritten, presenting the most relevant goal. It was highlighted the use of a statistical design for the optimization of CFD process, the use best conditions of both CFD and ozonation processes, the best results, including energy consumption from the O3/Fe2+/UV-C system.

  1. Also, I wouldn’t recommend the use of abbreviations in the abstract

Thank you for your very complete analysis. We removed the abbreviation for winery wastewater and ferrous iron. But to avoid repetition and make the summary too long we thought it would be convenient to abbreviate some words more common such as coagulation-flocculation-decantation (CFD) and total organic carbon (TOC).

3 In the introduction section, the authors should also highlight the advantage of ozone in the oxidation process in front of its competitors. For example, what could have been a disadvantage of using chlorine in the oxidation process in place of ozone?

Thank you for your help, your observations improve this work. In page 2, line 69 – 74 it was pointed out the advantages of using ozone in comparison to other oxidants such as H2O2, , Cl2, and O2. In addition, it was highlighted the advantages of using radiation in combination with ozone.

  1. The type of sampling and the number of samples used in the statistical analysis should be highlighted in the manuscript.

The number of samples used in the statistical analysis were written in page 7, line 223 and line 242.

  1. In table 1, the TSS has been listed but the analytical process of total suspended solids has not been described in the text.

In page 3, line 112 – 115, it was written the process for TSS determination along with turbidity, pH and conductivity.

  1. The experimental setup is not clear, as to how much wastewater goes to the system, an estimate of how much gets out as purified water, and an estimate of sludge. The hydraulic retention time for each sub-unit within the integrated system should be presented. Some information like power consumption of each sub-unit or the entire treatment plant could add a piece of more useful information.

Reviewer 3, thank you for your observations, it helps the authors to improve this work. First, regarding the observation “The experimental setup is not clear, as to how much wastewater goes to the system, an estimate of how much gets out as purified water, and an estimate of sludge”, it was not determined the sludge content after the CFD process, because the main objective was to determine the capacity of potassium caseinate and activated sodium bentonite to remove turbidity and TSS from the wastewater. However, in future works, this analysis will be performed. Regarding ozonation process, very little sludge is produced after oxidation of the wastewater, and the sludge volume can be disregarded.

Regarding the observation “The hydraulic retention time for each sub-unit within the integrated system should be presented”, as observed in Figure 2, the equipment used was very rudimentary and it was not possible to determine hydraulic retention time, however it will be performed in future works.

Regarding the observation “Some information like power consumption of each sub-unit or the entire treatment plant could add a piece of more useful information”, in page 14, line 435 – 454 and page 15, line 456 – 473, it was added information regarding energy consumption by the O3/Fe2+/UV-C system. In Equation 27, it was presented a figure of merit, Electric Energy per Order (EEO), which allows to understand the energy consumption in kWh m-3 order-1. The authors added Table 6, in which it is presented the EEO values from the different variations performed in the optimization of this work. In addition, authors added Table 7, in which it is presented some examples of energy consumption observed on other works which performed oxidation by ozonation process.

  1. The role of each sub-unit in the integrated treatment approach is not clear.

To understand the role of each sub-unit, initially in page 7, in material and methods, it was separated in two sub-section (2.5 and 2.6), in which the experimental conditions of CFD process and ozonation process were explained in detail. In addition, it was added Figure 2, which shows the set-up of ozonation reactor and how each part fits in the reactor.

In Table 1, it was showed that the winery wastewater has high levels of turbidity (1040 NTU) and TSS (2430 mg/L) and organic content (1962 mg C/L). Therefore, in page 8, line 250 – 255 (section 3.1) it was clarified that CFD process had the objective of reducing mainly the turbidity and TSS, since these substances can adsorb radiation and decrease the efficiency of UV driven processes.

In page 11, line 339 – 342 (section 3.2) it was defined that ozonation process had the main objective of reducing the organic carbon present in the winery wastewater.

In page 16, line 494 – 504 (section 3.3) it was defined that the main objective in this section was to evaluate the combination of the best operational conditions obtained in section 3.1 and 3.2 for CFD and ozonation processes.

The authors believe that the role of each sub-unit in the integrated approach is clear.

  1. The authors should also include latitude and longitudes of the case study, with a clear reference (for example: certain km from the capital city or international airport), as well as briefly present the nature of activities at the industry

The information that you have requested are presented in page 2, line 90 – 98.

  1. All analytical methods including test kits and reagents with the manufacturer’s name, city and country should be well presented.

The information that you have requested are presented in page 3, line 99 – 103 and line 105 – 122.

  1. It could have been more interesting if the authors also checked the performance of the treatment approach for microbial elimination.

Reviewer 3, thank you for your observations. Unfortunately, it was not possible for the authors to evaluate the performance of the treatment approach for microbial elimination. However, your advice will be taken in consideration for future works.

  1. Could also be interesting if the authors presented a schematic diagram of the treatment plant

In Figure 2, it is presented a schematic diagram of the ozonation reactor.

  1. All statistical methods used in the study with their respective tools should be highlighted in the materials and methods section

In page 7, line 220 – 221, it was presented in the material and methods the statistical method and software used to perform the CFD process was indicated. In addition, it is indicated the number of experiments performed and standard deviation.

  1. With the fact that the experimental setup has not been well presented, it becomes difficult to properly follow the results and discussion section.

Experimental setup was improved with addition of Figure 2. The authors believe that it is easy to understand the setup of the experimental process and the results obtained. In addition, section 2.5 and section 2.6 are well separated and each have a detailed information of the experimental conditions of each process used in the winery wastewater treatment.

  1. In most cases, you will find that achieving relatively high removal efficiencies is fine, but the removal efficiencies lead to effluent quality to be used for what purpose becomes the question. Therefore, a comparison with some recommended water quality standards or guidelines could provide a more useful piece of information.

In page 10, line 333 – 334 and page 17, line 518 – 522, it was added some recommended water quality guidelines for wastewater discharge.

  1. The quality of the figures should also be improved.

The quality of Figure 2 and Figure 3 were improved.

  1. I would suggest designing the conclusion section as one paragraph based on the objective, approach, and the results’ merits, work novelty, and future applicability.

The conclusions should reflect the main results being very difficult to put in a single paragraph. However, in the conclusion, it was added a sentence (page 20, line 620 – 623), reflecting the treatments approach, and the results’ merits, work novelty, and future applicability.

  1. In general, the paper should be thoroughly checked in terms of writing style, e.g., L24: “WW treatment can be an efficient treated”. This statement is not grammatically correct.

The document was reviewed and changes in grammar were performed.

Reviewer 4 Report

The manuscript deals with "treatment of winery wastewater with integration of coagulation and ozonation methods".

1. Write keywords alphabetically.

2. In the Materials and Methods, please draw a schematic of the integrated reactor.

3. If applicable, the kinetic study of ozonation, and ozone consumed (OC) should be done.

4. Quality of figure 2 should be improved.

Author Response

The manuscript deals with "treatment of winery wastewater with integration of coagulation and ozonation methods".

1. Write keywords alphabetically.

Reviewer 4, thank you for your advice, the keywords were written alphabetically.

2. In the Materials and Methods, please draw a schematic of the integrated reactor.

Reviewer 4, thank you for your review. A schematic illustration of the integrated reactor is in fact important to understand the system used for the winery wastewater treatment. Therefore, it was added by the authors the Figure 2, in which it was drawn a schematic of the integrated reactor.

3. If applicable, the kinetic study of ozonation, and ozone consumed (OC) should be done.

In page 12, line 385 – 395, it was indicated the pseudo-first order kinetic rate and half-life (Equation 21 and 22) of degradation of TOC by ozonation process. The kinetic values are also presented in Table 6.

4. Quality of figure 2 should be improved.

Reviewer 4, thank you for your advice, Figure 2 is now Figure 3 and its quality was improved.

Round 2

Reviewer 2 Report

I appreciate authors efforts to improve their manuscript. However, still some points should be considered properly before publication.  

1- Authors stated in page 10 line 313 that " Activated sodium bentonite has an isoelectric point of 7, becoming electronegative at pH 3.6 – 4.0". Here in, I raise several issues; a) Please measure the actual pH of the bentonite used in your study (Do not provide values from literature as you use different clay mineral).b) You almost miswritten that clause again, for any chemical structure when pH<pHiep, this structure will be positively charged indeed. c) According to the pH of the bentonite suspension used in your study and in presence of PVPP, please provide a possible yet detailed mechanism for the removal process of target pollutants.

2- The manuscript's language needs extensive refining, PLEASE, consider as a substantial issue. 

Author Response

"I appreciate authors efforts to improve their manuscript. However, still some points should be considered properly before publication.  

1- Authors stated in page 10 line 313 that " Activated sodium bentonite has an isoelectric point of 7, becoming electronegative at pH 3.6 – 4.0". Here in, I raise several issues;

  1. a) Please measure the actual pH of the bentonite used in your study (Do not provide values from literature as you use different clay mineral)."

Reviewer 2, thank you for addressing this issue. The authors measured the pH of the Bentonite (pH = 7.4). The pH was indicated in page 10, line 313-316.

  1. b) "You almost miswritten that clause again, for any chemical structure when pH<pHiep, this structure will be positively charged indeed."

Reviewer 2, the sentence was corrected as it can be observed in page 10, line 313-316.

  1. c) "According to the pH of the bentonite suspension used in your study and in presence of PVPP, please provide a possible yet detailed mechanism for the removal process of target pollutants."

Reviewer 2, in page 10, line 327-340 the authors wrote a possible mechanism that occurred when bentonite, potassium caseinate and PVPP were added to the winery wastewater. In addition, in page 11, Figure 4 was added, showing the action of the different coagulants.

"2- The manuscript's language needs extensive refining, PLEASE, consider as a substantial issue."

An extensive review of English language was performed by the authors. In accordance with the recommendation, we carefully reviewed the entire text of the manuscript.

Reviewer 3 Report

The authors have done impressive work to address most of the comments, and the manuscript is way far better.

Here are a few more comments:

  • L107: The term “biological oxygen demand” should be changed to “biochemical oxygen demand”.
  • The borders in equation 2 should be removed.
  • Language should again be checked, for example, L574: “Considering the low studies regarding….”, Here, I think the authors wanted to portray that, there are no enough studies that have investigated the impact of CFD and ozonation processes in the phenolic composition of the WW; check the use of the term “low”, and is always good if those kinds of claims are justified.
  • L602: “It was successfully used potassium caseinate…..”, the statement should be rephrased.
  • The authors should also try to highlight all the software used in the statistical analysis with the version numbers.

Otherwise, the authors have done tremendous work in improving the manuscript, and I would recommend it for publication after addressing the minor issues.

Author Response

“The authors have done impressive work to address most of the comments, and the manuscript is way far better.

 Here are a few more comments:”

  1. “L107: The term “biological oxygen demand” should be changed to “biochemical oxygen demand”.”

Reviewer 3, thank you for your help. In page 3, line 106, it was written “biochemical oxygen demand”.

  1. “The borders in equation 2 should be removed.”

The borders in Equation 2 were removed.

  1. “Language should again be checked, for example, L574: “Considering the low studies regarding….”, Here, I think the authors wanted to portray that, there are no enough studies that have investigated the impact of CFD and ozonation processes in the phenolic composition of the WW; check the use of the term “low”, and is always good if those kinds of claims are justified.”

Reviewer 3, in page 20, line 590-598, the sentence was justified properly. The justification for performing these analyses were due to the fact that these compounds exist in the wine and are transferred to the winery wastewater. There is a lack of information however regarding how these types of treatments affect the degradation of these compounds.

The term “low” was replaced, and the sentence was improved.

  1. “L602: “It was successfully used potassium caseinate…..”, the statement should be rephrased.”

In Page 14, line 622-626, the sentence was rephrased.

  1. “The authors should also try to highlight all the software used in the statistical analysis with the version numbers.”

Reviewer 3, in Page 7, line 219, line 225 and line 245, it was highlighted the software used to perform the statistical analysis.

“Otherwise, the authors have done tremendous work in improving the manuscript, and I would recommend it for publication after addressing the minor issues.”

Reviewer 4 Report

Reviewers’ comments have been addressed

Author Response

“Reviewers’ comments have been addressed.”

Thank you very much for your work.